# Structures of a sperm-specific solute carrier gated by voltage and cAMP

Valeria Kalienkova[1,3], Martin F. Peter[1,2], Jan Rheinberger[2] & Cristina Paulino[1,2 ✉]

The newly characterized sperm-specific Na$^+$/H$^+$ exchanger stands out by its unique tripartite domain composition[1,2]. It unites a classical solute carrier unit with regulatory domains usually found in ion channels, namely, a voltage-sensing domain and a cyclic-nucleotide binding domain[1,3], which makes it a mechanistic chimera and a secondary-active transporter activated strictly by membrane voltage. Our structures of the sea urchin SpSLC9C1 in the absence and presence of ligands reveal the overall domain arrangement and new structural coupling elements. They allow us to propose a gating model, where movements in the voltage sensor indirectly cause the release of the exchanging unit from a locked state through long-distance allosteric effects transmitted by the newly characterized coupling helices. We further propose that modulation by its ligand cyclic AMP occurs by means of disruption of the cytosolic dimer interface, which lowers the energy barrier for S4 movements in the voltage-sensing domain. As SLC9C1 members have been shown to be essential for male fertility, including in mammals[2,4,5], our structure represents a potential new platform for the development of new on-demand contraceptives.

Solute carriers (SLCs) constitute the largest class of membrane transporters with more than 450 members. They show a high genetic, functional and structural variability and are increasingly acknowledged as an untapped source of potential new drug targets[6–9]. One example is the sperm-specific SLC9C1 that belongs to the SLC9 superfamily of cation/proton antiporters, known as Na$^+$/H$^+$ exchangers (NHEs), which are essential for the regulation of intracellular pH, sodium homeostasis and cell volume[10]. SLC9C1 was shown to be essential for male fertility and its expression shown to directly correlate with sperm count and motility[2,4,5,11].

SLC9C1 sets itself apart from other SLCs by its unique domain composition, which is not found in any other membrane protein described to date. The transport domain (TD) is accompanied by a voltage-sensing domain (VSD) and a cyclic-nucleotide binding domain (CNBD), which are regulatory units usually found in voltage-gated cyclic-nucleotide-modulated ion channels, also referred to as CNBD ion channels[1,3]. In a recent patch-clamp fluorimetry study, the functional integrity of the domains was confirmed for SLC9C1 from the sea urchin *Strongylocentrotus purpuratus*, termed SpSLC9C1, identifying it as a Na$^+$/H$^+$ exchanger that is activated strictly by hyperpolarization and modulated by cyclic nucleotides (cNMPs)[1]. As such, SpSLC9C1 could be described as a 'molecular lego', where independently evolved functional units are combined to accommodate different environmental and cellular needs. Specifically, SpSLC9C1 adapted to mediate cellular alkalization in sperm only in response to chemoattractant-induced hyperpolarization.

This raises several intriguing mechanistic questions, as active transporters such as SLCs have evolved substantially distinct transport and regulation mechanisms compared with ion channels. NHEs obey an elevator-like alternating access mechanism, which entails a continuous back-and-forth cycling of the transporting unit between the two membrane leaflets[12–18]. A central aspect of this model is that substrates and counter-ions themselves gate the conformational change. In contrast, ion channels form a pore, which, when open, does not undergo large conformational changes during ion conduction. In voltage-gated ion channels (VGICs), the voltage sensor in the VSD moves upon changes in the membrane electric field, thereby either directly dilating the pore or releasing an obstruction downstream of the pore, which allows passive ion flux[19–23]. It is thus puzzling how membrane voltage sensed by a VSD can activate ion exchange in a secondary-active transporter. In this study, we determined structures of SpSLC9C1 in ligand-free and ligand-bound conformations that provide insights into the overall domain architecture of the SLC9C group of NHEs, and we reveal how the three functional domains might be coupled.

## Domain structure and coupling helices

We determined structures of the previously functionally characterized SpSLC9C1 (ref. 1), in the absence of its ligand in detergent and in lipid nanodiscs (Fig. 1, Extended Data Figs. 1–3 and Extended Data Table 1). The dataset obtained in detergent showed only a poorly resolved density for the membrane-embedded VSD located at the periphery of the molecule, with a subclass adopting an unusual 'tilted' conformation (Extended Data Fig. 3e,f). In contrast, all membrane-embedded domains were better resolved when reconstituted into nanodiscs, yielding a map at 3.2 Å that was used for subsequent model building (Fig. 1a and Extended Data Figs. 1e,f and 2a). We could identify concentric layers of lipids (Extended Data Fig. 4a) between the TD and VSD, which supports the idea that the structure and function of SLC9C1 might be strongly affected by the environment.

[1]Groningen Biomolecular Sciences and Biotechnology, University of Groningen, Groningen, The Netherlands. [2]Biochemistry Center, Heidelberg University, Heidelberg, Germany. [3]Present address: Department of Biomedicine, University of Bergen, Bergen, Norway. ✉e-mail: cristina.paulino@bzh.uni-heidelberg.de

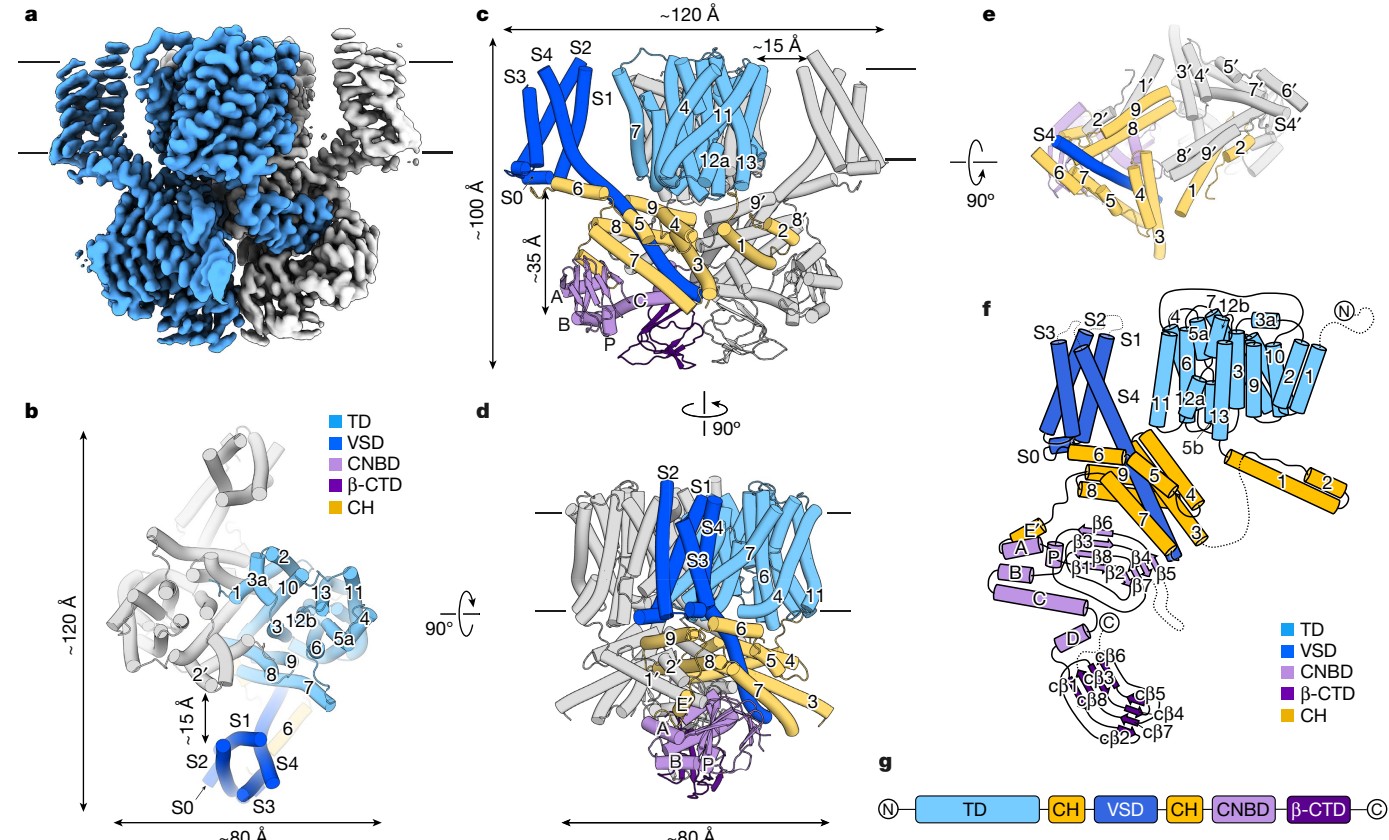

**Fig. 1 | Architecture of SpSLC9C1. a**, Cryo-EM map of the SpSLC9C1 dimer in nanodiscs, in the absence of ligands at pH 7.6 and 150 mM Na$^+$. Protomers are displayed in unique colours and membrane boundaries are indicated by horizontal lines. The map is contoured at 0.4σ. **b–d**, Model of SpSLC9C1 in a ligand-free conformation displayed as cylinders viewed from top (**b**), as in panel **a** (**c**) and from the side, rotated (**d**). **e**, The arrangement of the cytoplasmic helices viewed from top; transmembrane part is not displayed. Selected helices are labelled and the relationship between views is indicated. Individual domains are displayed in unique colours used throughout the manuscript unless otherwise indicated, namely, the TD in light blue, the VSD in dark blue, the CHs in yellow, the CNBD in light purple and the β-CTD in dark purple. **f**, Planar schematic representation of the SpSLC9C1 architecture. Dashed lines indicate loops neither resolved nor modelled. **g**, Domain arrangement of SpSLC9C1 on a sequence level.

Similarly to other NHEs, SpSLC9C1 assembles as a homodimer and the structures allowed the unambiguous identification of distinct domains, namely, a membrane-embedded TD, comprising 13 helices (TM1-13); a membrane-embedded VSD, consisting of helices termed S1–S4; and a soluble cytoplasmic domain (CTD), harbouring the CNBD (Fig. 1). Interestingly, the CNBD constitutes only a small fraction of the CTD, with the remainder composed of helices compactly arranged in a rhomboid shape (Fig. 1e). We tentatively termed them coupling helices (CH1–9), as they appear to form the only interactions between the three functional units—TD, VSD and CNBD. Further, they form a cytoplasmic dimer interface and interprotomer interactions, with helices CH1 and CH2 of one protomer wedged between the helix pair CH8′–CH9′ and the CNBD of the adjacent protomer (Fig. 1c–f). An approximately 30 amino-acids-long linker between CH2 and CH3 is not resolved and is presumably flexible. Helix pairs CH3–CH4, CH5–CH7 and CH8–CH9 form 'a scaffold' around the extended cytoplasmic part of S4 of the VSD. Helices CH6 and S0 are amphiphilic and positioned below the lipid bilayer boundaries (Fig. 1c–f and Extended Data Fig. 4b).

Several striking features set SpSLC9C1 apart from canonical CNBD channels[3]. Firstly, the VSD in ion channels is encoded directly upstream of the pore domain (PD), which ensures a direct coupling between the voltage sensor (S4) movements and the opening and closing of the pore (Extended Data Fig. 4c,d). In contrast, the VSD in SLC9C1 is encoded downstream of the TD, with no direct covalent connection between the S4 and the catalytic transport domain (Fig. 1, Extended Data Fig. 4c,d and Supplementary Fig. 2). In fact, the VSD is located at the periphery of the protein, laterally to the TD at a substantial distance (approximately 15 Å) from the exchanging unit (Fig. 1b,c). Secondly, the voltage-sensing helix S4 is substantially longer compared with any other known VSD structures and is embedded within the network of coupling helices, which may mediate the interaction between the VSD and TD (Fig. 1c–f and Extended Data Fig. 4c). Another difference is the sequential arrangement of the CNBD to the VSD and TD. In contrast to ion channels, the CNBD of SpSLC9C1 does not follow the catalytic transport regions, but is encoded downstream of the VSD, separated by CH7–CH9 (Fig. 1c,f,g, Extended Data Fig. 4c,d and Supplementary Fig. 2). Finally, another β-roll domain is found C-terminally to the CNBD, which we termed β-CTD. It displays features of the CNBD fold, but lacks the C-helix and the conserved phosphate binding cassette required to accommodate cyclic nucleotides (cNMP) (Extended Data Figs. 1f, 4c,d and Supplementary Fig. 2). Together with the CHs, this domain forms a cytoplasmic dimer interface (Fig. 1c and Extended Data Fig. 1g), in addition to the NHE-like dimer interactions found in the membrane-embedded TD. These structural elements show resemblance to the analogous cytosolic domains found in the recent structure of the plant NHE SOS1 (ref. 24).

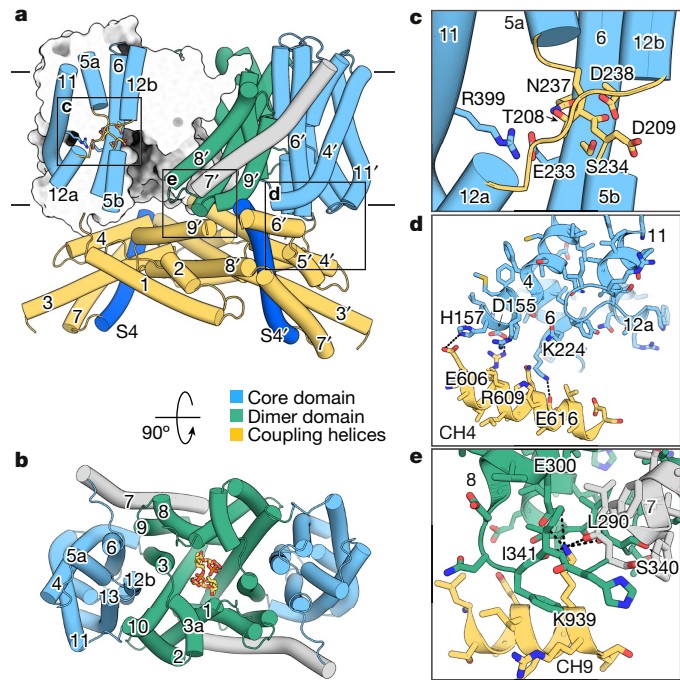

**Fig. 2 | Structural features of the SpSLC9C1 transport domain. a**, Secondary structure of the transport domain of SpSLC9C1 shown as cylinders. Core and dimer domains are displayed in light blue and green, respectively, connecting TM7 is in grey, coupling helices of the CTD are in yellow and the extended cytoplasmic region of the VSD helix S4 is in dark blue. The left protomer is represented as a slice-through surface, revealing an inward-facing state. Helices forming the ion-binding site are displayed, with the unwound crossing region of TM5 and TM12 shown as yellow loops. Residues important for ion coordination and activity are displayed as sticks, including D209 and the backbone carbonyl of T208 in the unwound part of TM5, D238 and S234 on TM6, and R441 on TM12, as well as the conserved salt bridge between E233 on TM6 and R399 on TM11. Membrane boundaries are indicated. **b**, Top view of the transport domain. Lipids at the dimer interface are displayed as orange sticks. **c**–**e**, Close-up views of regions boxed in **a**, highlighting the ion-binding site (**c**), potential interactions between the core domain of TD and the CTD (**d**) and the interface between the dimer domain of TD and the CTD (**e**).

## The TD shows a conserved NHE fold

The TD of SpSLC9C1 displays a typical NhaA fold[10,12], with a six transmembrane helix (TM) inverted repeat topology (Fig. 2). Although predicted to consist of 14 TMs[1], we could only identify 13 resolved helices per protomer, with the first 70 N-terminal residues unresolved (Fig. 2a,b and Supplementary Fig. 2). The nomenclature adopted for NHE transporters designates TM1–3 and TM8–10 as the dimer domain that forms the stable scaffold of the protein. TM4–6 and TM10–13 comprise the catalytic six-helix bundle motif, termed the core domain, which shuttles between the two membrane leaflets transporting the substrates by means of an elevator mechanism[13,17,18]. Helices TM5 and TM12 cross each other and are unwound around the conserved cation-binding site in the middle of the membrane, which, in SpSLC9C1, bears all important residues for ion coordination, including the 'ND' motif characteristic of electroneutral NHEs (Fig. 2a,c and Supplementary Fig. 2)[1,15,25,26]. The similarity between the TD fold and other NHEs is highlighted in the root mean square deviation (r.m.s.d.) between protomer structures around 2–3 Å (Extended Data Fig. 4e). The r.m.s.d. is somewhat higher on a dimer level, which reflects a higher degree of architectural freedom and overall mobility at the dimer interface, as reported for NHA2 (ref. 16) and hence might vary depending on the exact conditions and environment. Equally, it might highlight the impact that diverse cytoplasmic domains can have on the dimerization interface and its dynamics[24,26,27].

SpSLC9C1 adopts an inward-facing conformation, similar to the majority of eukaryotic NHE structures[15,26,27], with the residues of the binding site accessible from the cytoplasm forming a negatively charged cavity (Fig. 2a,c and Extended Data Fig. 4f). Several non-protein densities were resolved in the cryo-electron microscopy (cryo-EM) map, which are largely attributed to bound lipid molecules. Prominent examples are the clear densities corresponding to two lipids embedded in the cavity found at the extracellular half of the dimer interface of the TD (Fig. 2b and Extended Data Fig. 4a,f,g). The densities did not allow us to unambiguously identify which lipid is bound, although it resembles an emerging common feature seen for other NHEs[15,16,27].

## The VSD exhibits an extended S4

The VSD of SpSLC9C1 shows a typical fold, resembling that found in VGICs, with S1–S4 helices arranged as a four-helix bundle within the membrane (Fig. 3a–c and Extended Data Fig. 5a,b). Compared with hyperpolarization-activated and cNMP-modulated ion channels[28–30], SpSLC9C1 has a longer S1–S2 linker, which appears mobile and is unresolved in the cryo-EM density maps (Extended Data Fig. 5a and Supplementary Fig. 2). In contrast, the S2–S3 linker is short and wrapped around by the CH6 and S0 linker. Whereas helix S3 in hyperpolarization-dependent channel VSDs is kinked at two positions[28–30], it is relatively straight in SpSLC9C1 (Fig. 3b,c and Extended Data Fig. 5a,b). The VSD of SpSLC9C1 resembles that of the depolarization-activated Shaker Kv channel[31], including a short amphipathic helix S0 (Extended Data Fig. 5b). SpSLC9C1 has seven positively charged residues in a canonical $(R/K–XX)_n$ pattern on S4, which almost matches the number of those in HCN1 (ref. 28) and HCN4 (ref. 29) (Fig. 3c and Extended Data Fig. 5a). The voltage-sensing helix S4 is in the inactive upward position, as no activating hyperpolarizing conditions can be established in detergent or nanodiscs. All positively charged residues reside within the membrane and $R_1$ (R803), which is homologous to R368 in Shaker Kv and is supposed to cross the entire membrane electric field upon activation[1,32–34], is still high above the gating charge transfer centre (GCTC) (Fig. 3c). Interestingly, the GCTC of SpSLC9C1 and that of other SLC9C1 homologues, has a Tyr rather than a Phe, as found in the majority of VSDs[35,36] (Extended Data Fig. 5a and Supplementary Fig. 2). Yet, the VSD of SpSLC9C1 was shown to be a functional voltage sensor[1], in line with extensive mutagenesis studies on the Shaker Kv channel, which concluded that only Tyr and Trp substitutions preserved the wild-type-like channel activation at more negative membrane potential[36].

The most remarkable feature of the SpSLC9C1 VSD is the substantially elongated S4 helix, in which the soluble C-terminal part is approximately 56 Å long, protrudes into the cytoplasm by approximately 35 Å and is embedded within the CHs (Figs. 1c–f and 3b and Extended Data Figs. 4c and 5c–f). It is likely to serve as a 'stem' that supports the rest of the VSD within the membrane and possibly communicates the conformational transition of S4 through the coupling helices to the transport domain. The interactions between S4 and the coupling helices appear to be mediated by numerous hydrophobic residues (Extended Data Fig. 5d–f). Additional stabilization of the VSD in the membrane might be provided by amphiphilic helices CH6 and S0 (Extended Data Fig. 4b).

## The CNBD shows a different arrangement

In voltage-gated CNBD channels, the CNBD usually follows the S6 pore helix connected by a C-linker (CL) (Fig. 3a and Extended Data Fig. 4c,d). In SpSLC9C1, although the CH8–CH9 helix pair shows some resemblance to a C-linker, the overall structural similarity between the CNBDs of VGICs and that of SpSLC9C1 begins only with helix E' (Fig. 3a,b,d and Extended Data Figs. 5c and 6a). The orientation of

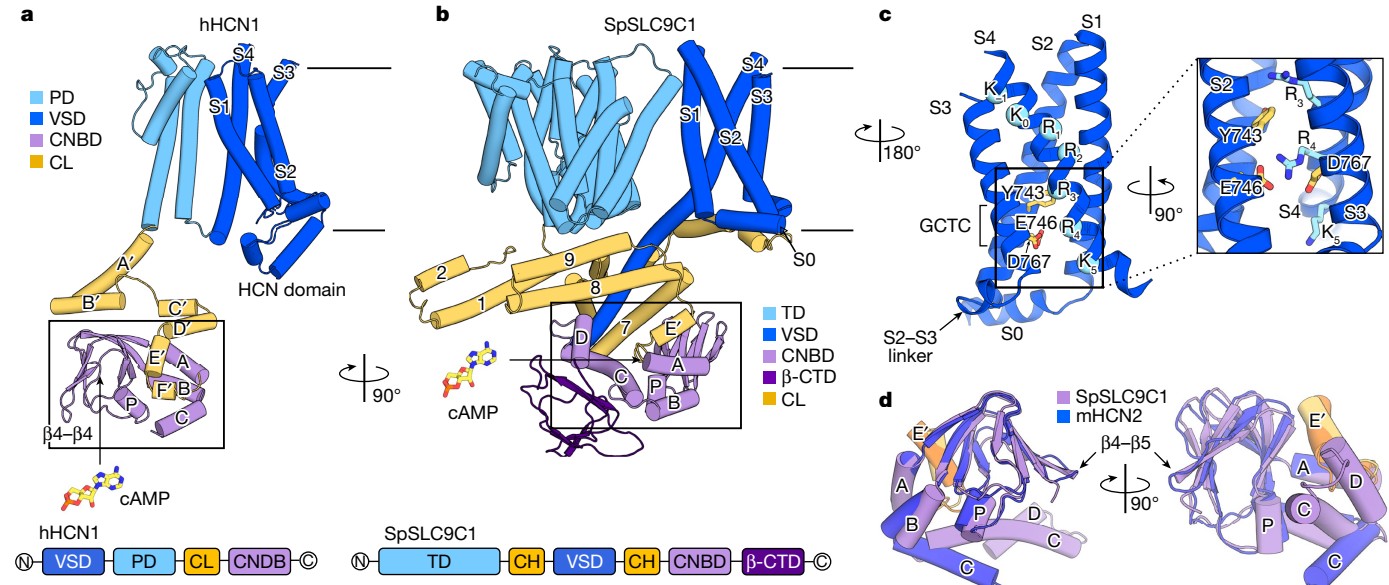

**Fig. 3 | The VSD and CNBD of SpSLC9C1 in comparison with canonical CNBD channels. a,b,** General architecture of hHCN1 (PDB: 5U6O) (**a**) and SpSLC9C1 (**b**). Top, structural comparison of the two proteins. A single protomer is displayed for clarity. Equivalent functional domains are coloured accordingly: VSD in dark blue; TD and PD in light blue; CL and CH in yellow; CNBD in light purple; and β-CTD in dark purple. The relationship between boxed regions is indicated by rotation. Bottom, primary sequence domain arrangement of hHCN1 and SpSLC9C1. **c,** Close-up of the membrane-embedded part of the VSD of

SpSLC9C1. Gating charge transfer centre (GCTC, composed of Y743, E746 and D767) on S2–3 is indicated. Conserved positively charged residues are displayed as blue spheres and labelled according to the Shaker Kv nomenclature. Right, close-up of the GCTC. Conserved positively charged residues are shown as blue sticks and GCTC residues are shown as yellow sticks. **d,** Overlay of the SpSLC9C1 CNBD (yellow and light purple) with that of mHCN2 (PDB: 5JON, orange and dark purple). Selected structural elements of CNBDs are labelled.

the β-CNBD with respect to the rest of the protein is also different in SpSLC9C1. Whereas in ion channels it is located under the C-linker with the cNMP-binding site facing outwards, it is rotated clockwise by 90° in SpSLC9C1 if viewed from the top (Fig. 3a,b). Consequently, the nucleotide binding site is facing the symmetry axis of the molecule (Fig. 3b) and helices C and D come into direct contact with the cytoplasmic tip of S4 of the VSD (Extended Data Fig. 5f). Additionally, the C-helix, which together with the β-CNBD and the phosphate binding cassette forms the nucleotide binding site, is less mobile and, consequently, better resolved, as well as being longer and positioned substantially closer to the binding site compared with ligand-free structures of various CNBD channels[28,29,37,38] (Fig. 3d and Extended Data Fig. 6b). Finally, the loop between β4 and β5 of the β-CNBD is considerably longer in SLC9C1 homologues compared with CNBDs of hyperpolarization-activated channels. However, it is not resolved in the density (Extended Data Figs. 1h and 6a).

## cAMP induces conformational changes

Cyclic nucleotides were shown to facilitate the activation of SpSLC9C1 by shifting the activation voltage closer to resting membrane potential, with a stronger effect detected for cAMP compared with cGMP, whereby only cAMP caused a detectable shift in the onset of transport[1]. To investigate the impact of ligand binding on the conformation of SpSLC9C1, we solved the structures of the protein in the presence of cGMP and cAMP (Fig. 4, Extended Data Figs. 7 and 8 and Extended Data Table 2). To resolve the structural heterogeneity, we performed focused classifications and refinements on a dimer and protomer level for both ligand-bound datasets. In all cases, we could identify prominent densities within the CNBD corresponding to bound nucleotides, which were not present in the apo cryo-EM maps (Extended Data Figs. 1h, 7g and 8g). Overall, the best-resolved cGMP- and cAMP-bound dimer structures, at 3.2 Å and 3.3 Å, respectively, share

a similar conformation to the apo structure (Extended Data Fig. 8h). However, we could identify key differences. Although image processing of the cGMP dataset revealed only one dominant conformation, the cAMP-bound dataset displayed a high degree of conformational heterogeneity throughout data analysis, and, in particular, the CTD was generally less resolved and mobile (Fig. 4a and Extended Data Figs. 7d,e and 8b,d,e,i). In most of the three-dimensional (3D) classes and as observed in the 3D variability analysis, the dimer interaction between the β-CTDs is disrupted in the presence of cAMP, which causes the CTD to swing away from the symmetry axis (Fig. 4a, Extended Data Fig. 8d,i and Supplementary Video 1). The observed flexibility in the CTD is also accompanied by a higher mobility of the VSDs, although at no point does the VSD approach the transport domain. Further, the extracellular tip of TM1 rotates around P75 towards the symmetry axis, pushing the TD protomers apart (Extended Data Fig. 6e). The observed movement is similar to the 'breathing motions' described for NHE9 (ref. 15), which underlines the dynamic nature of the TD dimer interface as a common feature in the SLC9 family. On a protomer level, two distinct classes could be resolved that largely resemble conformational changes observed on a dimer level and which are best appreciated in a morph between both cryo-EM maps (Extended Data Fig. 8d–f,j and Supplementary Video 2). Sufficiently resolved regions further allow us to describe in more detail the conformational heterogeneity induced by cAMP binding. Here, we observed pronounced movements of, in particular, CH3 and CH4, as well as CH7 and S4 upwards, away from the CNBD, and the downward movement of the CNBD, best resolved for helix C (Fig. 4d and Supplementary Video 2).

In canonical CNBD channels, the C-helix was shown to undergo most of the ligand-induced conformational changes (Fig. 4b and Extended Data Fig. 6c)[28,29,37,39]. In contrast, the CNBDs seen in all three dimeric SpSLC9C1 states overlay remarkably well, irrespective of the presence of ligand (Fig. 4c and Extended Data Fig. 6c). Minor conformational

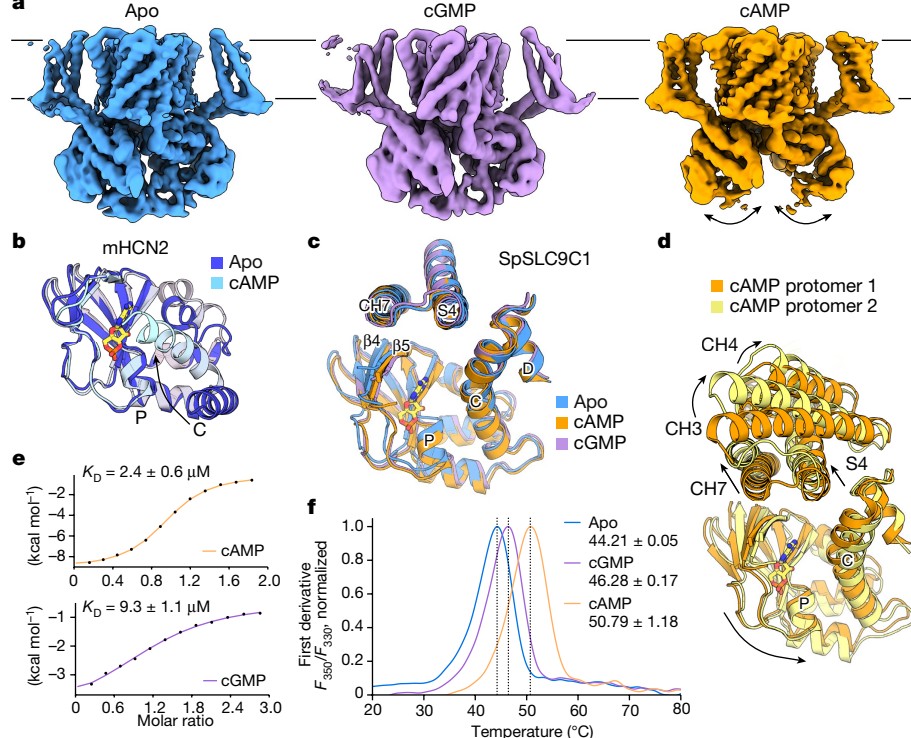

**Fig. 4 | Conformational changes associated with ligand binding in SpSLC9C1. a,** Intermediate refined unsharpened maps of SpSLC9C1 in apo (displayed at 4σ), cGMP-bound (4.2σ) and cAMP-bound (5.2σ) states illustrating the higher mobility of the CTD in the presence of cAMP. **b,** Overlay of the CNBD structure of mHCN2 in apo (dark blue, PDB: 5JON) and cAMP-bound (cyan, PDB: 3BPZ) conformations. The C and P helices are labelled, the movement of the former upon ligand binding is indicated by an arrow and cAMP is shown as a stick. **c,** Overlay of the CNBD structure of SpSLC9C1 observed in the dimeric apo (blue), cGMP (purple) and best-resolved cAMP-bound class (orange) conformations. Selected helices are indicated and cAMP is shown as a stick. **d,** Overlay of the two cAMP-bound protomer classes obtained through extensive 3D classification which discloses stronger movements within the CNBD as well as CHs, as seen for CH3, CH4 and CH7. **e,** ITC binding curves for the isolated SpSLC9C1–CTD construct (S946–E1193) titrated with cAMP (top) or cGMP (bottom). The mean binding affinity and s.d. for three biological replicates are given. **f,** nanoDSF measurement of the isolated SpSLC9C1–CTD construct for three technical replicates. The normalized first derivates of the ratio of the detected fluorescence signals at 350 nm ($F_{350}$) and 330 nm ($F_{330}$) for the apo protein (blue) and after addition of cGMP (purple) and cAMP (orange) are shown.

differences between the three structures are β4 and β5 of the β-CNBD positioning closer to the bound nucleotides and a slight movement of helix C towards the β-CNBD in the cAMP-bound structure. However, it is questionable whether the resolved cAMP-bound state represents a 'fully activated' conformation. Firstly, a ligand-induced movement of the C-helix analogous to that seen in CNBDs of VGIC that 'closes' the cNMP-binding site would result in a clash with the extended cytoplasmic part of S4 (Fig. 4b,c). Secondly, we observe a high degree of conformational heterogeneity in the CTDs upon cAMP addition. Therefore, we hypothesize that a full conformational transition of the C-helix upon cAMP binding will require further rearrangements within the cytoplasmic domain. This is supported by the presumably 'initial' transitions observed in the cAMP-bound protomer state 2 (Fig. 4d, Extended Data Fig. 8j and Supplementary Video 2). Whereas in the apo structures the C- and D- helices are in direct contact with S4 and might preclude its movement, upon cAMP binding, S4 and the surrounding CHs move upwards, closer to the membrane, and the C-helix moves downwards, further away from S4, possibly releasing the inhibition. As a result, the β-CNBD might be approaching the C-helix reminiscent of other ligand-bound CNBD structures.

## Ligand binding and coordination

cAMP and cGMP bind SpSLC9C1 in a similar fashion as reported for other cNMP-bound CNBD structures[38–41], adopting an 'anti' and a 'syn' conformation, respectively, with most ligand coordinating residues conserved (Extended Data Fig. 6a,d). Interestingly, unlike other ligand-bound VGIC–CNBD structures in which R1097 (equivalent in HCN2 R632) interacts with cNMP by means of its backbone carbonyl, in both of our ligand-bound structures the residue interacts by means of its sidechain, which is concomitant with the unusual position of the rather immobile helix C (Extended Data Fig. 6d).

To further characterize the effect of cGMP and cAMP binding on SpSLC9C1, we analysed the isolated CTD (residues S946–E1193, comprising helix E', the CNBD and the β-CTD), as carried out previously for HCN channels[42] (Extended Data Fig. 6a and Supplementary Fig. 2). Differential scanning fluorimetry (nanoDSF), as well as isothermal titration calorimetry (ITC), confirmed the functional integrity of the isolated domain (Fig. 4e,f and Extended Data Fig. 6f). The ITC measurements showed one clear binding event for each substrate. For cAMP, a four-fold higher affinity was measured (2.4 μM) compared with cGMP (9.3 μM), which is comparable to cNMP-binding studies of HCN channels[43,44] and consistent with cAMP being a strong agonist. Analogously, the nanoDSF measurements showed a ligand-induced increase in thermal stability of the isolated CTD, where cAMP produced a shift of 6 °C instead of only 2 °C observed for cGMP. The results corroborate the higher efficacy of cAMP for SpSLC9C1 potentiation. However, it is not trivial to explain this effect from a structural point of view, as recently described for the SthK channel[40], as most of the identified residues are not conserved in SpSLC9C1.

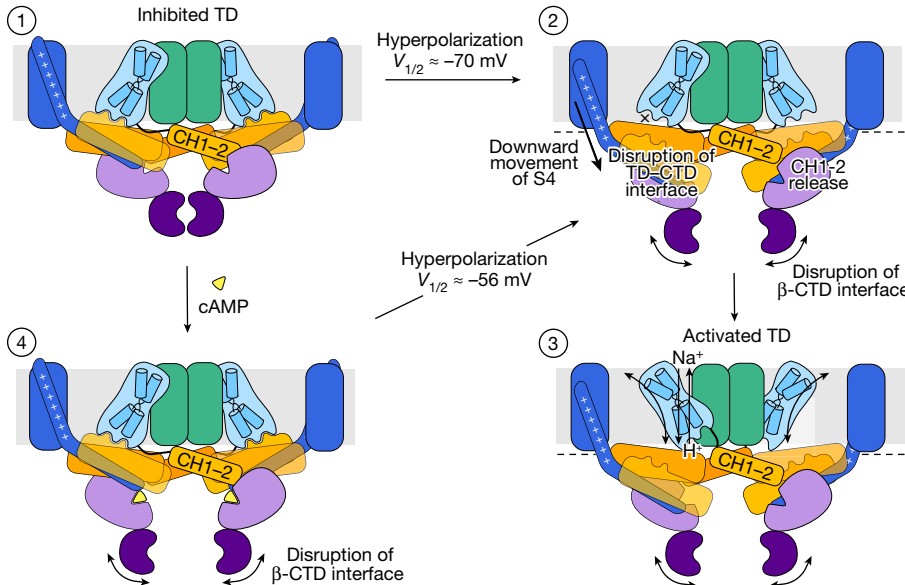

**Fig. 5 | Putative activation model for the voltage-gated and cAMP modulated Na⁺/H⁺ exchange in SpSLC9C1.** A schematic representation of the proposed SpSLC9C1 activation mechanism. Under non-activating conditions, the TD is locked in an inward-facing state through interactions with coupling helices (state 1). Membrane hyperpolarization causes a downward S4 movement, which presumably leads to the disruption of the interfaces between the TD and CTD and between adjacent β-CTDs. The increased mobility of the CTD would also release the interactions of the CH1–2 helices within the adjacent protomer

(state 2). The described changes unlock the arrested catalytic core domain of the TD, which allows Na⁺/H⁺ exchange by means of an elevator-like mechanism (state 3). Binding of the strong agonist cAMP increases the conformational dynamics of the CTD, thereby removing one of the barriers for activation (state 4). Ligand-free and various intermediate cAMP-bound states were observed in this study (state 1 and 4), whereas states 2 and 3 with S4 in a down conformation are postulated.

## Potential interdomain coupling

A general observation, evident from the local resolution maps (Extended Data Figs. 2a, 7e and 8e), is that in all of our SpSLC9C1 structures the protein displays substantial structural heterogeneity for most of its domains except for the TD, which indicates that it might be locked. Focused 3D classifications at the dimer level could not resolve the conformational flexibility, which suggests that the movements of the domains are, to a degree, independent within the dimer. Therefore, focused heterogeneity analysis at the protomer level with a mask excluding the TD was performed instead. For the apo dataset, four conformations with a resolution range of 3.2–3.4 Å and with substantially improved densities of all the domains could be distinguished (Extended Data Figs. 1e and 2b–f). Equally relevant are the two distinct protomer states found in the cAMP dataset, which show the highest degree of conformational changes (Fig. 4d, Extended Data Fig. 8j and Supplementary Video 2). The most pronounced differences are confined to the region of the CHs. They are accompanied by subtle conformational transitions in the VSD and CNBD, which further indicates that the CHs might play a central role in coupling voltage sensing and cAMP modulation to exchange activity in the TD. In line with this assumption, the extended cytoplasmic region of S4 forms several interaction areas with CHs, which have to rearrange upon voltage activation (Extended Data Fig. 5d,e). Further, we could identify a network of salt bridges between CH4 and the core domain of the TD (E606–H157, R609–D155, K224–E616), which might contribute to locking the transporter in the inward conformation (Fig. 2d). A contributing factor to this might be coupling helices CH1–2, which are directly encoded after the core domain and are embedded within the CTD helices of the neighbouring protomer and would have to move alongside the core domain during the elevator mechanism (Fig. 1c–e). Another interesting position is K939 on CH9, which is coordinated by E300, and backbone carbonyl groups located in the dimer domain (Fig. 2e). Because it does not interact with the mobile

core domain, we hypothesize that this region might be important for anchoring the CTD to the TD.

## Discussion

SpSLC9C1 is a remarkable chimeric protein that demonstrates the adaptable repertoire of nature's building blocks and mechanistic concepts. Our structures reveal how these conserved domains are arranged into a new fold, as seen in SpSLC9C1 (Fig. 1). Although, on the individual domain level, the TD, VSD and CNBD of SpSLC9C1 display conserved characteristics and folds, their overall arrangement starkly contrasts with that found in canonical CNBD channels. Most prominent are: (1) the absence of direct interactions between the VSD and the TD; (2) the new cytoplasmic coupling helices that link all three functional domains; and (3) the additional β-CTD, with the latter two forming a cytoplasmic dimer interface (Fig. 1). We confer the CHs a central role in the gating mechanism and envision a two-fold function. Firstly, under non-hyperpolarized (resting) conditions, they ensure that the TD is locked in an inactive state, by: (1) a network of salt bridges between CH4 and TD helices 4 and 6 (Fig. 2a,d); and (2) the helix pair CH1–2, which is presumably immobilized by helices CH8′–9′ and the CNBD of the neighbouring protomer (Fig. 1c–f). An outwardly directed elevator-movement of the TD during the transport cycle is likely to require the disruption of these interactions. Secondly, the CHs act as an allosteric transducer, by mediating conformational changes and thereby coupling all three functional units. The cAMP-bound structures provide a glimpse into the mechanism of modulation of SpSLC9C1 by cNMPs and putative coupling between the functional domains by means of the CHs. In general, movements of the CHs are linked to subtle rearrangements in the VSD and CNBD, and binding of its strong agonist cAMP induces large conformational transitions within the CTD (Fig. 4a, Extended Data Figs. 2f and 8j and Supplementary Videos 1 and 2).

Based on the structures obtained in this study, we propose a gating mechanism for SpSLC9C1 (Fig. 5). Under non-activation conditions, the protein is arrested in an inward-facing conformation due to interactions of the TD with the CTD (Fig. 5, state 1). As reported for other VGICs[45–47], it is expected that the S4 of the VSD moves downwards upon membrane hyperpolarization. Hereby, the unusually long cytosolic extension of S4 will displace several CHs, causing large-scale conformational rearrangements within the CTD. This leads to the disruption of interactions between the CTDs and between the CTD and the TD, which releases the exchanger unit from its arrested state (Fig. 5, state 2). When the lock on the TD is removed, SpSLC9C1 can facilitate Na$^+$/H$^+$ exchange in a classical SLC9-like fashion (Fig. 5, state 3). Lastly, cAMP binding alone is not sufficient to activate Na$^+$/H$^+$ exchange but has been shown to modulate the voltage dependency of SpSLC9C1, as manifested by a shift in $V_{1/2}$ (ref. 1). Here, binding of cAMP causes the disruption of the dimer interface in the CTD, conferring it a higher mobility (Fig. 5, state 4). We suggest that this removes one of the barriers that SpSLC9C1 has to overcome to be voltage activated, which allows an easier transition of S4 into its downward conformation.

In addition to SLC9C1 standing out as an intriguing phylogenetic and mechanistic chimera, it bears the potential for translational applications. SLC9C1 is essential for male fertility[2,4,5] and a comparative genomic analysis identified the molecular trio of SLC9C1, the soluble adenylyl cyclase sAC and the Ca$^{2+}$ channel CatSper as an evolutionary highly conserved machinery for the regulation of sperm flagellar beat in metazoans[48–50]. Hence, SLC9C1 is a potential target for the treatment of male infertility, as well as for non-hormonal on-demand male contraceptives, which is analogous to a recent strategy proposed for sAC[51]. However, the exact function of mammalian SLC9C1s in sperm capacitation remains unclear, as they lack conserved residues at the cation-binding site and mice SLC9C1$^{-/-}$ sperm display a defective cAMP signalling, rather than an impaired intracellular pH regulation[2,11,52–54]. These questions and the linked pharmacological potential would benefit from further characterization of mammalian SLC9C1, including how mammalian SLC9C1s differ from SpSLC9C1 functionally and structurally.

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

## Methods

### Cell lines

Suspension-adapted HEK293S (ATCC CRL-3022) cells were grown at 37 °C, 5% carbon dioxide, 70% humidity and 185 r.p.m. in TPP600 bioreactors in FreeStyle media supplemented with 1% FBS and antibiotic-antimycotic solution. Sf9 cells (ThermoFisher Scientific, 12659017) were cultivated at 27 °C, 130 r.p.m., in SFMIII supplemented with antibiotic–antimycotic. No further authentication of the cell lines was performed. Cultures were tested for mycoplasma contamination every 3–4 months and were found negative.

### Constructs

SpSLC9C1 gene (Uniprot-ID: A3RL54) codon-optimized for mammalian cells was synthesized by means of Genscript. The expression vector pEZT-BM was a gift from Ryan Hibbs (Addgene plasmid no. 74099)[55] and was adapted for FX-cloning[56] with a C-terminal HRV-3C cleavage site, followed by Venus, myc- and SBP-tags. Bacmid was generated following Invitrogen Bac-to-Bac protocol and isolated as described[57].

### Expression and purification of the full-length SpSLC9C1

For the full-length protein, the virus was generated as described[55]. Briefly, Sf9 cells were seeded to $1 \times 10^6$ cells per well into six-well plates and transfected with purified bacmid using Cellfectin II reagent (ThermoFisher Scientific Inc.). After 4–5 days, the supernatant was filtered and diluted 1:1000 into Sf9 cells in suspension at $1 \times 10^6$ cell density. Infection was monitored using GFP fluorescence. When the majority of cells were fluorescent, the virus-containing supernatant was filtered and stored at 4 °C with 2% FBS. One day before transduction, the HEK293S were diluted to a cell density of $0.6 \times 10^6$. The following day, the viral stock was diluted to 1:10 into the expression culture and cells were grown at 37 °C for another 24 h. Afterwards, the cells were supplemented with sodium butyrate (final concentration 10 mM) and transferred to 30 °C and grown for another 48–72 h. Cells were harvested by centrifuging at 550g, washed with ×1 PBS and stored at −80 °C until further use.

All purification steps were performed on ice or at 4 °C, unless stated otherwise. On the day of purification, cells were resuspended in buffer A (20 mM HEPES pH 7.6, 150 mM NaCl, 10% glycerol, 2% DDM, 0.4% CHS, DNAse I 100 µg ml⁻¹, 2 mM MgCl₂, cOmplete protease inhibitor tablets) and the protein was extracted for 2 h. The insoluble fraction was separated by centrifuging at 205 k$g$ in a Beckman Ti45 rotor for 30 min. Afterwards, the supernatant was applied to NHS-activated Sepharose with immobilized 3K1K[58] nanobody targeting the Venus tag on SpSLC9C1. The protein was bound to the beads in batch for 30 min and the supernatant was subsequently passed through a gravity-flow column three times. The resin was washed with approximately 50 column volumes of buffer B (20 mM HEPES pH 7.6, 150 mM NaCl, 10% glycerol, 0.02% GDN). For cryo-EM sample preparation in detergent, the protein was cleaved off in minimal volume using HRV-3C protease. The protein was concentrated using 100 kDa cut-off Amicon concentrators at 700 g and injected onto a Superose 6 Increase 10/300 column equilibrated in buffer C (20 mM HEPES pH 7.6, 150 mM NaCl, 0.02% GDN). The main peak fractions were concentrated as described above to 3.6 mg ml⁻¹ and used for sample preparation.

For nanodisc reconstitution, soy polar lipids (Avanti) were pooled, dried and washed with diethyl ether using a rotary evaporator. After the lipids were dried again to remove the solvent, they were rehydrated in nanodisc lipid buffer (20 mM HEPES pH 7.6, 150 mM NaCl, 30 mM DDM), at a final concentration of 10 mM. For sample preparation in nanodiscs, the protein was reconstituted while bound to the resin, with the molar ratio of SpSLC9C1 to membrane scaffold protein (MSP) 2N2 to lipids of 2:20:2,600, assuming ×10 excess of empty nanodiscs. Before assembly, the column was washed with several column volumes of buffer C. The assembly was performed at room temperature. Soy polar lipids were added to the resin with immobilized protein and the mixture was incubated for 30 min in batch. Afterwards, MSP was added and the protein was incubated in batch for another 30 min. Then, the biobeads were added (200 mg of beads per millilitre of the reaction) and the mixture was incubated overnight with gentle agitation. During the following purification steps, detergent was excluded from the buffers. The next day, the flowthrough containing empty MSPs was discarded and the resin was washed with approximately 5 column volumes of buffer C. The protein-containing nanodiscs were cleaved off using HRV-3C. The sample was concentrated using 100 kDa cut-off Amicon filter units at 500 g and injected onto Superose 6 equilibrated in buffer C. Fractions of the peak representing lower molecular weight nanodiscs were pooled and concentrated as above. For ligand-free and cGMP-bound datasets, the protein was concentrated to approximately 1 mg ml⁻¹, and, for preparing the cAMP-bound sample, the protein was concentrated to 11.8 mg ml⁻¹. Ligands (cAMP or cGMP) were added shortly before freezing to a final concentration of 2 mM. Additionally, the cAMP-bound sample was supplemented with fluorinated fos-choline 8 (final concentration 2.9 mM) to improve particle orientation on the grid.

### Cryo-EM sample preparation and data acquisition

Quantifoil Au 1.2/1.3 grids (mesh 300) were glow-discharged for 30 seconds at 5 mA before sample application. A volume of 2.8 µl of the purified sample was applied onto the grids, blotted for 3.5 seconds at 100% humidity, 15 °C, blot force 0, plunge-frozen in an ethane–propane mixture and stored in liquid nitrogen until data collection.

Ligand-free datasets of SpSLC9C1 in detergent and lipid nanodiscs were collected at the Dutch cryo-EM facility NeCEN, on a Titan Krios equipped with a K3 camera, postcolumn energy filter with 20 eV slit and a 100 µm objective aperture. Cryo-EM images were acquired in the automated fashion using AFIS implementation of EPU 2.7.0 or 2.8.0, at a nominal magnification of 105,000, with a calibrated pixel size of 0.836, 75 frames per image with a total dose of 60 e⁻/Å. The ligand-bound datasets were acquired at the University of Groningen on Talos Arctica equipped with a K2 Summit detector, postcolumn energy filter with 20 eV slit and 100 µm objective aperture. Cryo-EM images were recorded in an automated fashion using serialEM 3.8.0 beta or 3.9.0 beta[59] and a 3 × 3 multishot data acquisition pattern. The images were collected at a nominal magnification of 130,000, with a calibrated pixel size of 1.022, 60 frames per image with a total dose of 52 e⁻/Å. No statistical methods were used to predetermine the sample size. Cryo-EM data collection was deemed sufficient if it was possible to classify structural heterogeneity and to reach the resolution better than 4 Å in all of the reconstructions during subsequent image processing.

### Image processing

All the data were acquired using an in-house ice thickness estimation script[60] and preprocessed in Focus 1.1.0 (ref. 61). The data were motion corrected using MotionCor2 1.4.0 (ref. 62) and contrast transfer function (CTF) estimation was performed using ctffind4.1.14 (ref. 63). Cryo-EM images displaying ice contamination, resolution worse than 6 Å, and defocus values outside the range of 0.3–2.0 µm (Talos datasets) and 0.3–3.0 µm (Titan Krios datasets) were discarded. Particles were picked first crYOLO 1.7.5, 1.7.6 or 1.8.2 (ref. 64) with a self-trained model and extracted with a box size 300 (binned ×2 for ligand-free SpSLC9C1 dataset in detergent). For each dataset, two rounds of two-dimensional (2D) classification were performed in cryoSPARC v.3 (ref. 65), followed by ab initio reconstruction and heterogeneous refinement (usually on a smaller subset of particles) to generate an initial reference model. The particle sets obtained after 2D classification were imported into Relion 3.1.0 (ref. 66) and subjected to 3D classification for all datasets with the exception of cAMP-bound SpSLC9C1. For the latter, two consecutive rounds of ab initio reconstruction and heterogeneous refinement were performed to address the structural heterogeneity before importing particles from the best classes to Relion 3.1.0. In most cases,

one round of 3D classification in Relion 3.1.0 was sufficient to isolate high-resolution particles contributing to the final dimeric maps. These particles were subjected to several rounds of Bayesian polishing and CTF refinement before the final masked refinement (mask encompassing entire protein density). At this point, C2 symmetry was imposed. For cAMP-bound SpSLC9C1, it was necessary to subtract the nanodisc signal before the final masked 3D classification to improve class separation. Final 3D classification yielded 535,040 particles, which were subsequently refined with a mask and symmetry C2 and sharpened using deepEMhancer 20220530_cu10 (ref. 67). Particles after nanodisc signal subtraction and before final 3D classification (1,693,193 particles) were re-imported into cryoSPARC v.3 (ref. 65) and subjected to 3D variability analysis (3DVA)[68] with symmetry C1 and five components. Lastly, for all of the datasets, the particles contributing to final dimeric maps were symmetry expanded, followed by partial signal subtraction to focus on single protomers. The subtracted particles were refined and further classified with a mask excluding TD and no image alignment to identify classes with best-resolved VSD and CTD. The identified protomer classes were then refined with a mask and sharpened using deepEMhancer 20220530_cu10 (ref. 67) to yield the final monomeric maps. Resolution of all the maps was estimated using Relion 3.1.0 postprocessing, with a mask excluding nanodisc or detergent micelle density, according to the 0.143 Fourier shell correlation (FSC) cut-off criteria. Particles were randomised during refinement and resolution estimated between even and odd groups (gold standard FSC).

### Model building and refinement
DeepEMhancer maps were used for model building and figure preparation and the final refined unsharpened maps were used for model refinement. The ligand-free dimeric SpSLC9C1 (PDB: 8PCZ) was built using a homology model of NHE9 (PDB: 6Z3Y) and cAMP-dependent PKA (PDB: 3J4Q). Coupling helices were built de novo and lower-resolution regions (that is, VSD, β-CTD) were modelled using Alphafold2 v.2.3.2 prediction[69]. The obtained model was used as a starting point for all the other structures obtained in the present study. In all cases, models were iteratively adjusted manually in coot v.0.9.8.1 (ref. 70), refined in Isolde v.1.6.0 (ref. 71) followed by refinement in Phenix v.1.20.1-4487 (ref. 72). PyMOL v.2.5.5, ChimeraX v.1.6.1 (ref. 73) and Chimera v.1.17.3 (ref. 74) were used for structure visualization.

### Expression and purification of the isolated CTD
The gene of isolated SpSLC9C1–CTD (S946–E1193) was amplified from the full-length construct and cloned into the p7Xc3GH vector (AddGene plasmid no. 47066) which fuses the protein to a C-terminal HRV-3C cleavage site, followed by a GFP tag and a ×10 His-tag[56]. The plasmid was transformed in *E. coli* BL21 and expressed in LB-medium after induction with 1 mM IPTG overnight at 25 °C. The cells were harvested in the morning of the next day and the pellets were stored at −80 °C. If not mentioned otherwise, all of the following steps were performed at 4 °C. For purification, the cell pellet was resuspended in 30–40 ml (per litre expression volume) of buffer D (20 mM HEPES pH 7.6, 150 mM NaCl). The cells were lysed by sonication (amplitude 50%) with 5 s on-, 5 s off-pulses for 5 min and centrifuged for 30 min at 20,000 r.p.m. The supernatant was filtered with a 0.8 μm filter and incubated with NHS-activated Sepharose with immobilized 3K1K (ref. 58) nanobody for 30 min at 4 °C. The solution was transferred to a gravity column, the flowthrough was discarded and the beads were washed with 50 column volumes of buffer D. For cleavage, the resin was incubated in a minimal volume with HRV-3C protease for 2 h. The eluted fraction was collected and the resin was washed with 2 column volumes of buffer D. The fractions were combined and supplemented with imidazole to a final concentration of 20 mM. The solution was loaded on equilibrated Ni-resin to separate HRV-3C protease from SpSLC9C1–CTD. The flowthrough was collected, concentrated to 1 ml using a 10 kDa cut-off Amicon filter and loaded on a Superdex 200 Increase 10/300 column, equilibrated

in buffer D. The elution of protein was monitored by absorption at 280 nm and fractions containing protein were analysed by means of SDS-PAGE. Fractions with SpSLC9C1–CTD were pooled, concentrated and directly used for further analysis or frozen in liquid nitrogen and stored at −80 °C.

### nanoDSF measurements
nanoDSF measurements were performed on a Prometheus Panta (NanoTemper) using the corresponding software for data collection and analysis (Panta.Control v.1.4.3 and Panta.Analysis v.1.4.3). The concentration of SpSLC9C1–CTD was adjusted to 1 mg ml$^{-1}$ using buffer D and was supplemented with 264 μM of cAMP or cGMP. Standard capillaries (NanoTemper) were filled with protein solution and directly loaded into the device. The temperature range for measurements was set to 20–90 °C with a temperature ramp of 1.5 °C per minute. All measurements were performed in three technical replicates and the software was used to automatically calculate the mean value and error. No statistical methods were used to predetermine the sample size.

### ITC measurements
ITC measurements were performed on a MicroCal PEAQ-ITC (Malvern Panalytical) using the corresponding software for experimental design, data collection and analysis (MicroCal PEAQ-ITC Control v.1.41 and MicroCal PEAQ-ITC Analysis Software v.1.41). The concentration of SpSLC9C1–CTD was adjusted to 20–30 μM and the final concentration was determined using ultraviolet/visible absorption at 280 nm. Based on the protein concentration, a substrate solution of cAMP or cGMP was prepared with a ×10 higher concentration (200–300 μM) using protein buffer D. Then, 300 μl of the protein solution was loaded into the measuring cell and 60 μl of the substrate solution was loaded into the syringe of the MicroCal PEAQ-ITC. The temperature of the measuring cell was set to 25 °C and the substrate was injected in 12 injections of 3 μl. All measurements were repeated with three biological replicates and for the final binding affinity the mean value and standard deviation (s.d.) from these triplicates was calculated. No statistical methods were used to predetermine the sample size.

### Reporting summary
Further information on research design is available in the Nature Portfolio Reporting Summary linked to this article.

## Data availability
Cryo-EM density maps, half maps and masks have been deposited in the Electron Microscopy Data Bank (EMDB). Atomic models are available through the Protein Data Bank (PDB). The data are available under the following accession codes: detergent-solubilized apo SpSLC9C1 symmetric class (EMDB: 17603) and asymmetric class (EMDB: 17604); nanodisc reconstituted apo SpSLC9C1 dimeric (PDB: 8PCZ, EMDB: 17596); apo SpSLC9C1 protomer state 1 (PDB: 8PD2, EMDB: 17598); apo SpSLC9C1 protomer state 2 (PDB: 8PD3, EMDB: 17599); apo SpSLC9C1 protomer state 3 (PDB: 8PD5, EMDB: 17601); apo SpSLC9C1 protomer state 4 (PDB: 8PD7, EMDB: 17602); cGMP-bound dimeric (PDB: 8PDU, EMDB: 17621); cGMP-bound protomer (PDB: 8PDV, EMDB: 17622); cAMP-bound dimeric (PDB: 8PD8, EMDB: 17605); cAMP-bound protomer state 1 (PDB: 8PD9, EMDB: 17607); and cAMP-bound protomer state 2 (EMDB: 17625). Cryo-EM images were deposited to the Electron Microscopy Public Image Archive (EMPIAR)[75] under the following accession codes: 11628 (ligand-free SpSLC9C1 in nanodiscs); 11629 (ligand-free SpSLC9C1 in detergent); 11635 (cAMP-bound SpSLC9C1 in lipid nanodiscs); and 11630 (cGMP-bound SpSLC9C1 in lipid nanodiscs). The following amino acid sequences were used for sequence alignments and are available from Uniprot: SpSLC9C1, NP_001091927.1; spotted gar, XP_015193550.1; salmon, XP_013979929.1; green sea turtle, XP_027674929.1; tiger snake, XP_026524154.1; mouse, NP_932774.3;

and human, NP_898884.1. The following protein structures were used in this study and are available in the PDB: 5U6O, 5JON, 3BPZ, 4CZB, 6Z3Z, 7P1K, 6V1X, 7SIP, 8PD2, 7NP3, 6CJQ, 5U6P, 7NP4, 6Z3Y and 3J4Q.

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

**Acknowledgements** We thank M. Punter and M. Stuart for maintenance of the image processing cluster. We also thank the EM facility and the lab at the University of Groningen, as well as P. Grijpstra and G. van den Bogaart for providing access and assistance with the mammalian cell facility at the University of Groningen. We thank J. Baßler, S. Griesel, U. Göbel and L. Nücker for their help in setting up the lab at the Heidelberg University. We further thank J. Kamenz for advice and access to the insect cell culture, C. Alvadia, J. Walter, K. Ramanadane and C. Thangaratnarajah for their advice on protein expression and purification, I. Sinning for access to the Prometheus Panta, and B. Poolman, D. Slotboom, the entire Enzymology group and all members of the Paulino lab for helpful discussions. This work benefited from access to the Netherlands Centre for Electron Nanoscopy (NeCEN) at Leiden University, an Instruct-ERIC centre, which was funded by the Netherlands Electron Microscopy Infrastructure (NEMI), project number 184.034.014 of the National Roadmap for Large-Scale Research Infrastructure of the Dutch Research Council (NWO). V.K. acknowledges funding from the Swiss National Science Foundation: Early.Postdoc Mobility fellowship P2ZHP3_187679 and Postdoc Mobility fellowship P500PB_203053. C.P. acknowledges funding from the Dutch Research Council: Nederlandse Organisatie voor Wetenschappelijk Onderzoek (NWO) Veni Grant 722.017.001 and NWO Start-Up Grant 40.018.016.

**Author contributions** V.K., M.F.P. and C.P. designed the research. V.K. performed full-length protein cloning, protein expressions and purifications, nanodisc reconstitutions, acquired all cryo-EM data and determined all structures. J.R. assisted V.K. with acquiring data at the Netherlands Centre for Electron Nanoscopy (NeCEN) and at the University of Groningen. M.F.P. cloned the truncated SpSLC9C1 constructs, expressed and purified the protein and performed ITC and nanoDSF measurements. All authors analysed the data. V.K., M.F.P. and C.P. prepared the manuscript.

**Competing interests** The authors declare no competing interests.

**Additional information**
**Correspondence and requests for materials** should be addressed to Cristina Paulino.

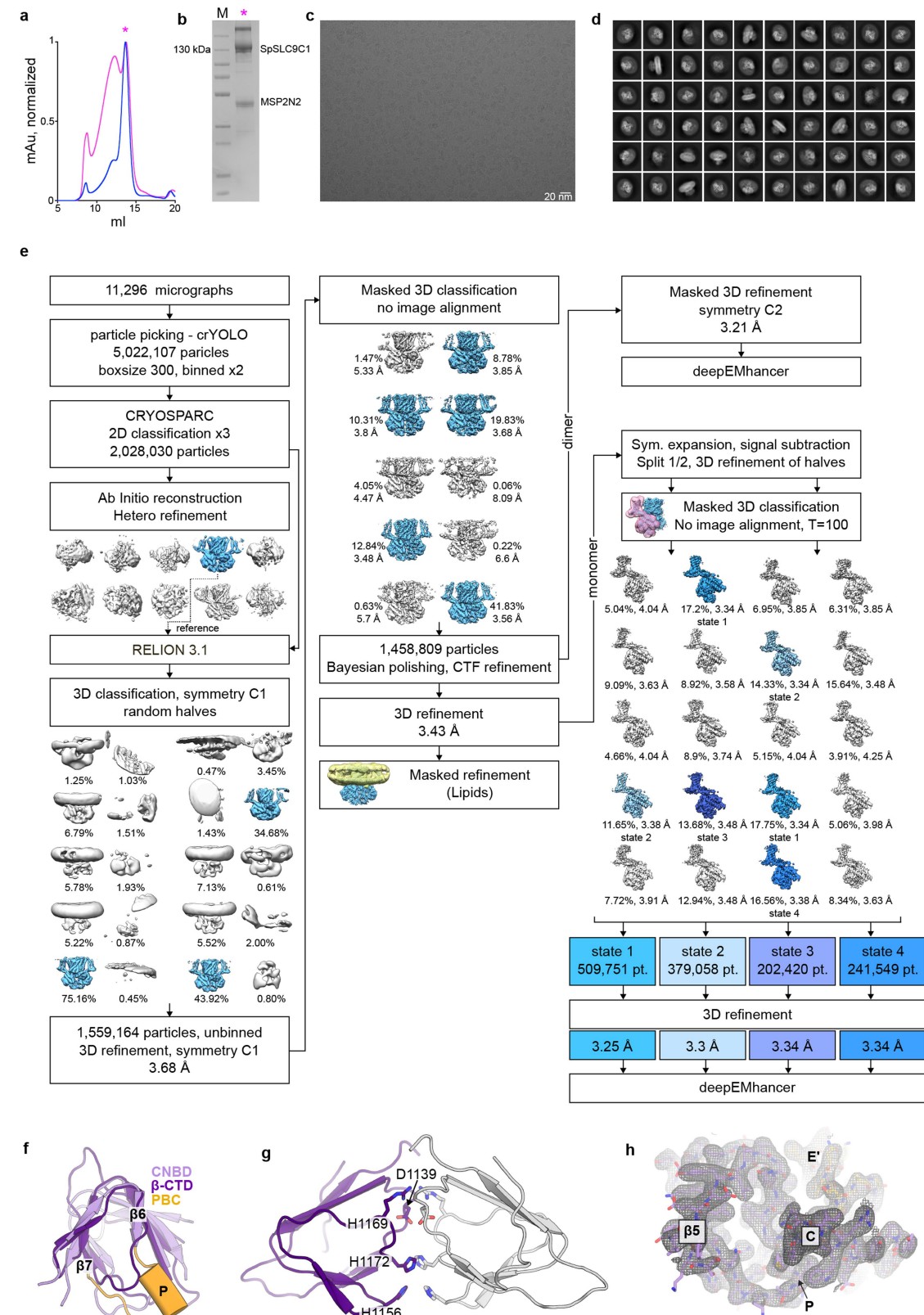

**Extended Data Fig. 1** | See next page for caption.

**Extended Data Fig. 1 | Cryo-EM of ligand-free SpSLC9C1 in lipid nanodiscs.**
a) Size exclusion profiles of detergent-solubilized (blue) and 2N2-reconstituted
(magenta) SpSLC9C1. Samples were analysed on Superose 6 Increase 10/300
column. b) SDS-PAGE of main peak fractions of SpSLC9C1 in lipid nanodiscs
used for grid preparation. Protein purification and reconstitution was
performed >3 times with similar appearance of size exclusion chromatograms,
and similar migration behaviour in the SDS-PAGE gel. For gel source data, see
Supplementary Fig. 1. Representative cryo-EM images out of 11,296 comparable
images (c), 2D classes (d) and image processing workflow (e). During 3D
classification of protomers, the mask encompassing only the VSD and CTD was
applied to focus on less-resolved parts of the protein. f) Overlay of the CNBD
(light purple) and the β-CTD (dark purple). The phosphate binding cassette
(PBC) important for cNMP coordination found in β-CNBD and not in β-CTD is
displayed in orange. g) Cytosolic inter-protomer interface of SpSLC9C1
mediated by β-CTD, selected residues are shown as sticks and labelled. As the
local resolution of the cryo-EM map does not allow an unambiguous modelling
of side chain rotamers in the β-CTD, a fit of the Alphafold model into the density
was used instead. It reveals a number of charged residues possibly interacting
between neighbouring β-CTD. h) cNMP-binding site in the CNBD, cryo-EM
density map is displayed as grey mesh contoured at 4.7 σ, SpSLC9C1 model is
shown in purple.

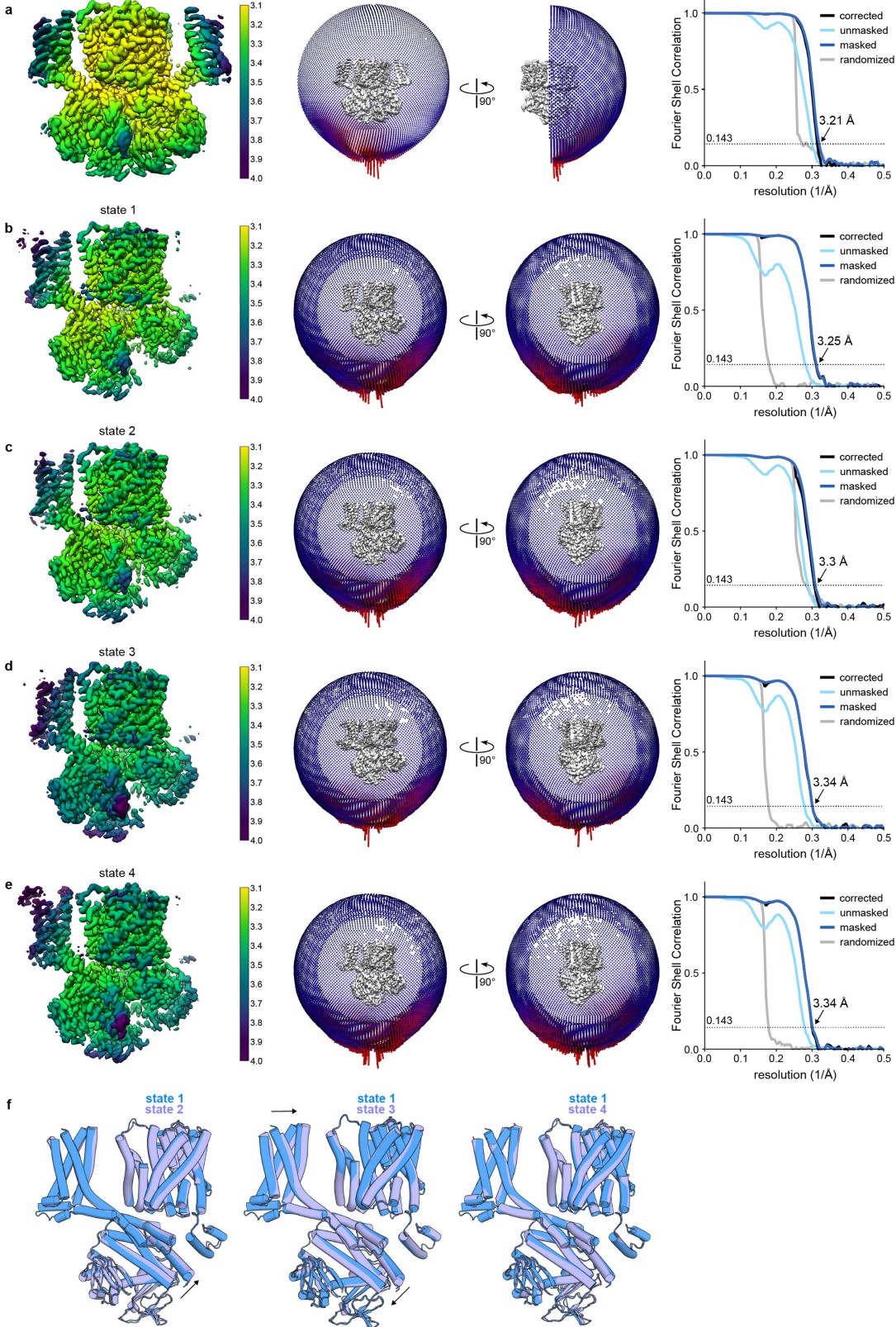

**Extended Data Fig. 2 | Validation of the ligand-free SpSLC9C1 in lipid nanodiscs dataset.** Shown are local resolution, angular distribution and Fourier shell correlation plots (0.143 criteria) of the SpSLC9C1 final dimer map, displayed at 0.4 σ (a), as well as of the protomer state 1, displayed at 1.3 σ (b), state 2, displayed at 1.6 σ (c), state 3, displayed at 0.6 σ (d) and state 4, displayed at 0.8 σ (e). f) Superposition of models obtained from the four distinct apo protomer states.

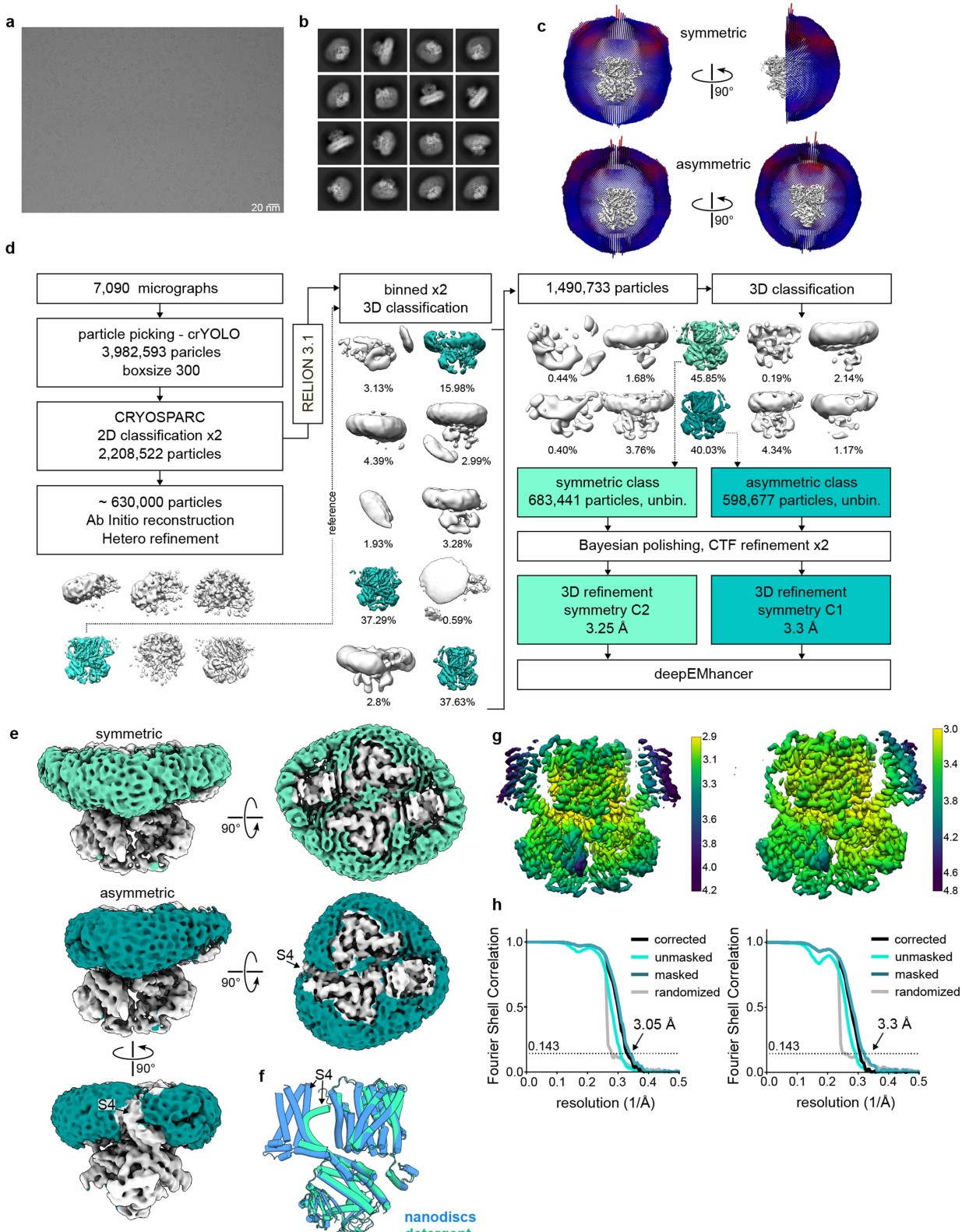

**Extended Data Fig. 3 | Cryo-EM of ligand-free SpSLC9C1 in detergent.** Representative cryo-EM images out of 7,090 comparable images (a) and 2D classes (b). c) Angular distribution of obtained final maps. d) Image processing workflow. e) Effect of the S4 conformation on micelle shape as seen in the obtained symmetric and asymmetric classes. Shown are refined unmasked maps lowpass-filtered to 5 Å. Protein is shown in grey, micelle in color. Asymmetric map is displayed at 3.8 σ, symmetric map at 3.5 σ. f) Overlay of the protomer state 1 obtained in nanodiscs and the asymmetric class in detergent,

S4 is indicated. It is likely that the detergent micelle is not sufficiently rigid to stabilize VSDs in the upright position, in contrast to lipid bilayer disc provided by the nanodisc. Therefore, we believe that the tilted/collapsed S4 conformation observed in a fraction of particles in detergent is unlikely physiological. We chose to not deposit the model and only use it for illustrative purposes. g) Final deepEMhancer sharpened maps colored according to the local resolution and displayed at 0.7 σ. h) Fourier-shell correlation (0.143 criteria) of the final maps.

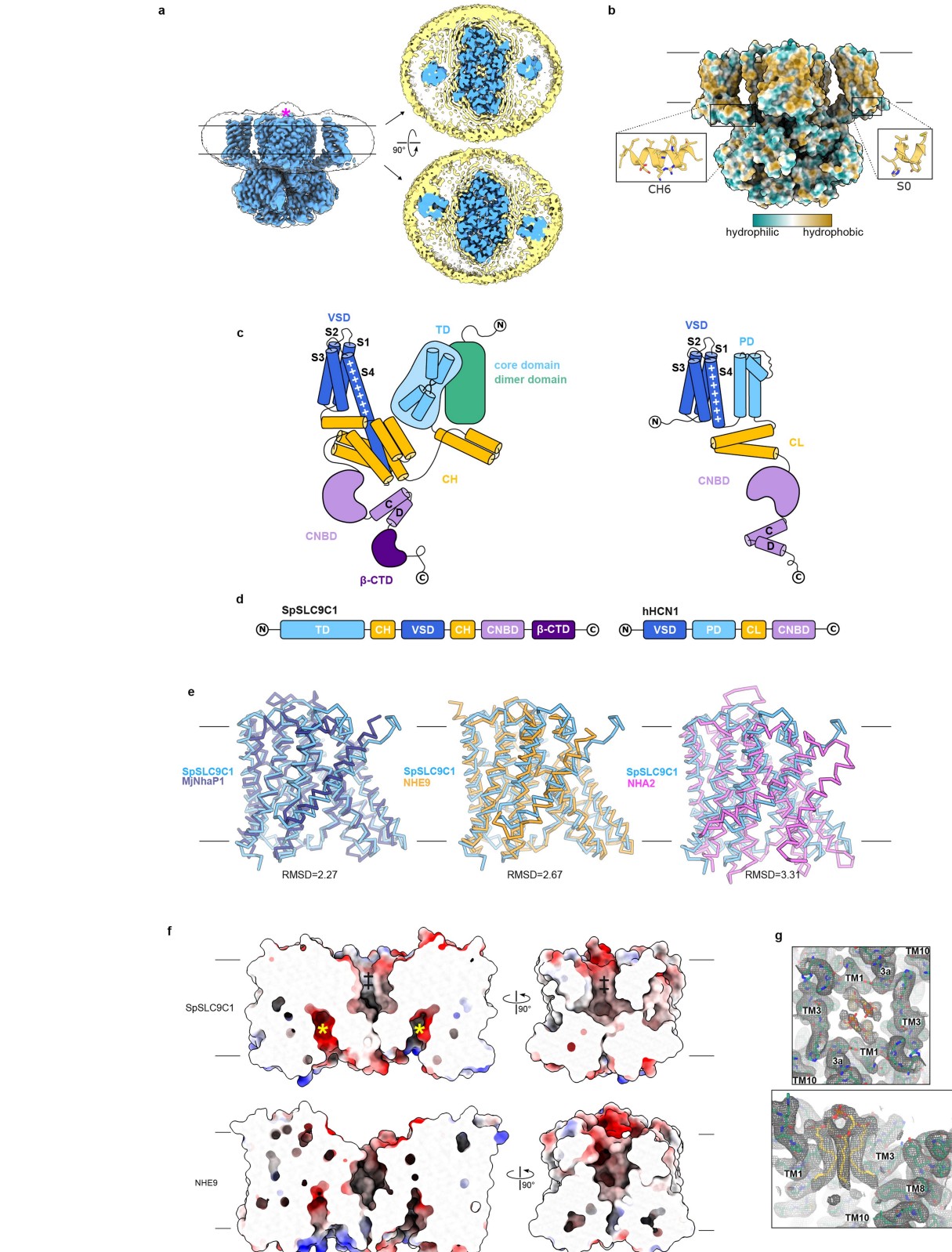

**Extended Data Fig. 4 | Lipid bilayer interactions and comparison to other NHEs.** a) Left, SpSLC9C1 apo map (blue), displayed at 0.4 σ overlayed with the unmasked refined map displayed at 2 σ (transparent) to indicate lipid bilayer boundaries. (*) marks a "cap" observed in the unmasked dimer map at a lower contour above the middle of TD, which might potentially be formed by the unresolved N-terminal region (residues 1-70). Right, slice-through of the SpSLC9C1 map (positions indicated on the left) refined with a mask encompassing the nanodisc region only, protein density is displayed in blue, surrounding lipid densities and MSP belt in yellow, map is contoured at 3 σ. b) Surface representation of apo SpSLC9C1 colored by hydrophobicity, membrane boundary is indicated. Amphipathic helices CH6 and S0 are displayed as cartoon. c) Schematic architecture of SLC9C1 in comparison to CNBD channels. Domains fulfilling equivalent functions are color-coded according to SpSLC9C1 – pore domain (PD) in light blue, VSD in dark blue, C-linker (CL) in yellow, CNBD in light purple. d) Primary sequence domain arrangement of SLC9C1 compared to CNBD channels, coloring as in (c). e) Overlay of the TD protomer of SpSLC9C1 with that of the archaeal MjNhaP1 (PDB: 4CZB), horse NHE9 (PDB: 6Z3Z) and bison NHA2 (PDB: 7P1K), r.m.s.d. values were calculated in coot using SSM superpose function and are indicated below individual overlays. f) Surface depiction of the TD colored by electrostatics of SpSLC9C1 and NHE9. Left, slice-through displaying the negatively charged cavity providing access to the ion-binding site (*) from the cytoplasm in the inward-facing conformation. Right, slice-through of the dimer interface (‡), showing the lipid filled extracellularly facing cavity (lipid densities not displayed). g) Top, close-up of the membrane-embedded dimer interface viewed from the top, protein is displayed in green, lipid molecules in orange. Selected helices are labelled. Refined unsharpened map is shown as grey mesh contoured at 4.7 σ. Bottom, view of the lipid densities from the side, one of the SpSLC9C1 protomers is not displayed for clarity.

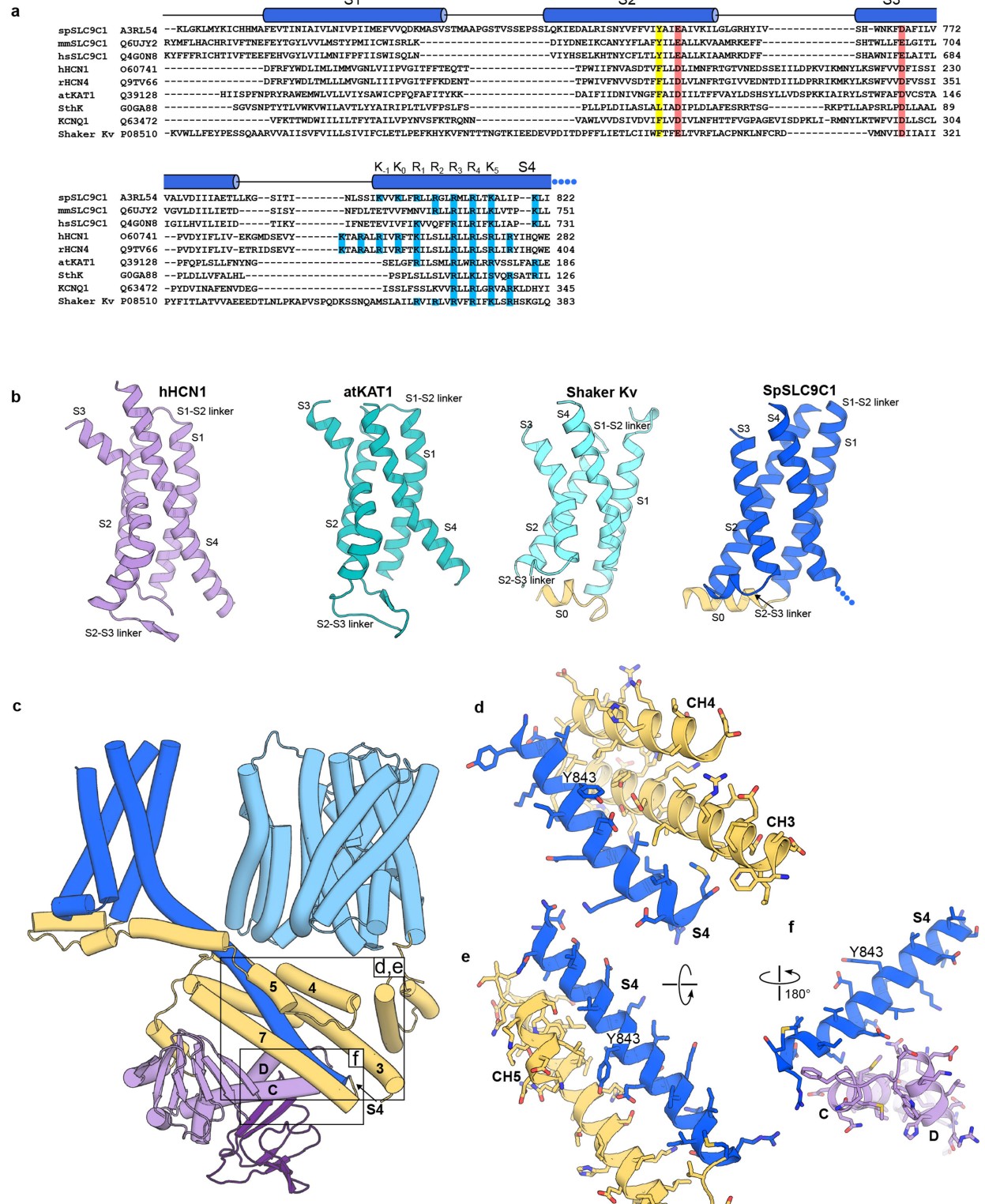

**Extended Data Fig. 5 | Comparison of the SpSLC9C1 VSD to CNBD ion channels and potential interdomain interactions.** a) Sequence alignment of SpSLC9C1 VSD to that of selected VGICs. GCTC residues are highlighted in yellow and red, gating charges in blue. Gating charges are numbered according to Shaker Kv. b) VSD structures of selected VGICs in comparison to SpSLC9C1; models used are from the hyperpolarization-activated hHCN1 (PDB: 5U6O) and atKAT1 (PDB: 6V1X), and the depolarization-activated Shaker Kv (PDB: 7SIP),

apo SpSLC9C1 protomer state 1 (PDB: 8PD2). c) SpSLC9C1 protomer state 1 (PDB: 8PD2) colored by domains, selected helices are labelled, and close-ups displayed in d)–f) are indicated. Interaction of cytoplasmic part of S4 with CH3, CH4 (d), CH7 (e) and CNBD helices C and D (f). The extended cytosolic region of S4 in SpSLC9C1 is indicated by blue dots. Y843 is displayed to facilitate the navigation between the panels.

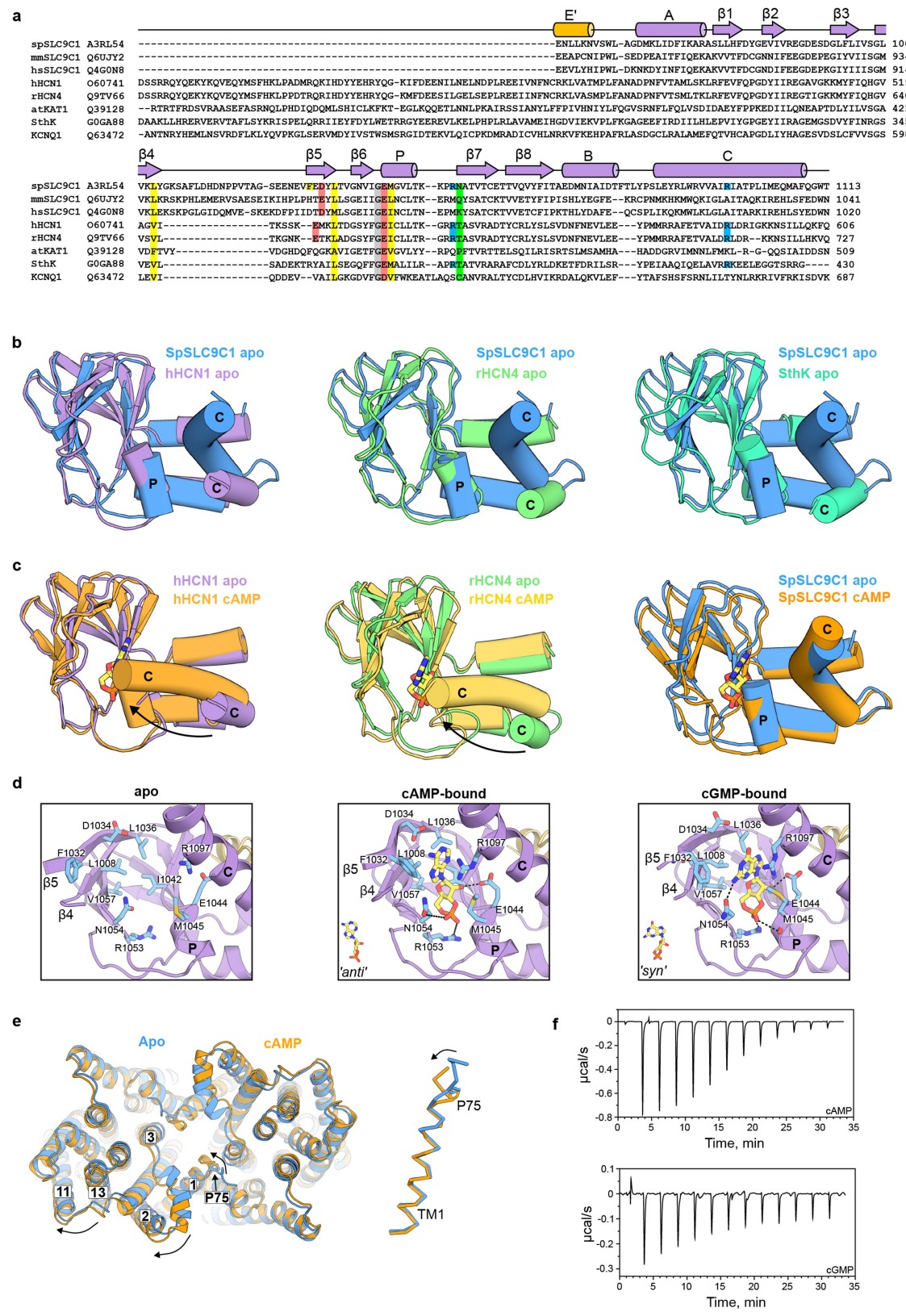

**Extended Data Fig. 6** | See next page for caption.

**Extended Data Fig. 6 | Comparison of the SpSLC9C1 CNBD, ligand binding and ligand-induced conformational changes.** a) Sequence alignment of the CNBD region of selected SLC9C1 homologs and CNBD channels. Residues interacting with cNMPs are highlighted according to their properties (hydrophobic in yellow, basic in blue, acidic in red, conserved glycine in grey. While HCN1 and SthK harbor cNMP-modulated CNBDs, KAT1 and KCNQ1 are cNMP insensitive as their CNBD do not carry the otherwise conserved arginine (R1053 in SpSLC9C1) that coordinates the cNMP phosphate group and is crucial for cNMP binding. b) Superposition of apo SpSLC9C1 protomer state 1 (PDB: 8PD2) with apo hHCN1 (PDB: 5U6O), rabbit HCN4 (PDB: 7NP3) and SthK (PDB: 6CJQ). Helices P and C of CNBDs are labelled. c) Superposition of CNBDs in apo (PDB-IDs as in (b)) and the respective cAMP-bound conformations (hHCN1 PDB: 5U6P, rHCN4 PDB: 7NP4, SpSLC9C1 PDB: 8PDV). P and C helices are labelled, conformational change observed in hHCN1 and rHCN4 upon cAMP binding is indicated by an arrow. d). Close-up of the cNMP-binding site observed for the CNBD in the apo (left), cAMP-bound (middle) and cGMP-bound (right) states, with cNMP molecules shown in yellow. Residues potentially important for cNMP coordination are shown in blue and labelled, these correspond to the conserved: L1008 on β4, F1032 and L1036 on β5, I1042 on β6, E1044 and M1045 on P-helix, R1053 on PBC and R1097 on C-helix. N1054 is stabilizing cGMP in a 'syn' conformation, equivalent to a Thr in HCN1, HCN2, HCN4 and SthK. e) Overlay of the transport domain of the apo (blue) and the best resolved cAMP-bound dimeric classes (orange), viewed from the top (left). Close-up of the TM1 in the two structures, the proline hinge is indicated (right). f) Representative ITC measurements for the isolated SpSLC9C1-CTD construct (S946-E1193) titrated with cAMP (upper panel) or cGMP (lower panel). The corresponding binding curves are shown in main text Fig. 4e.

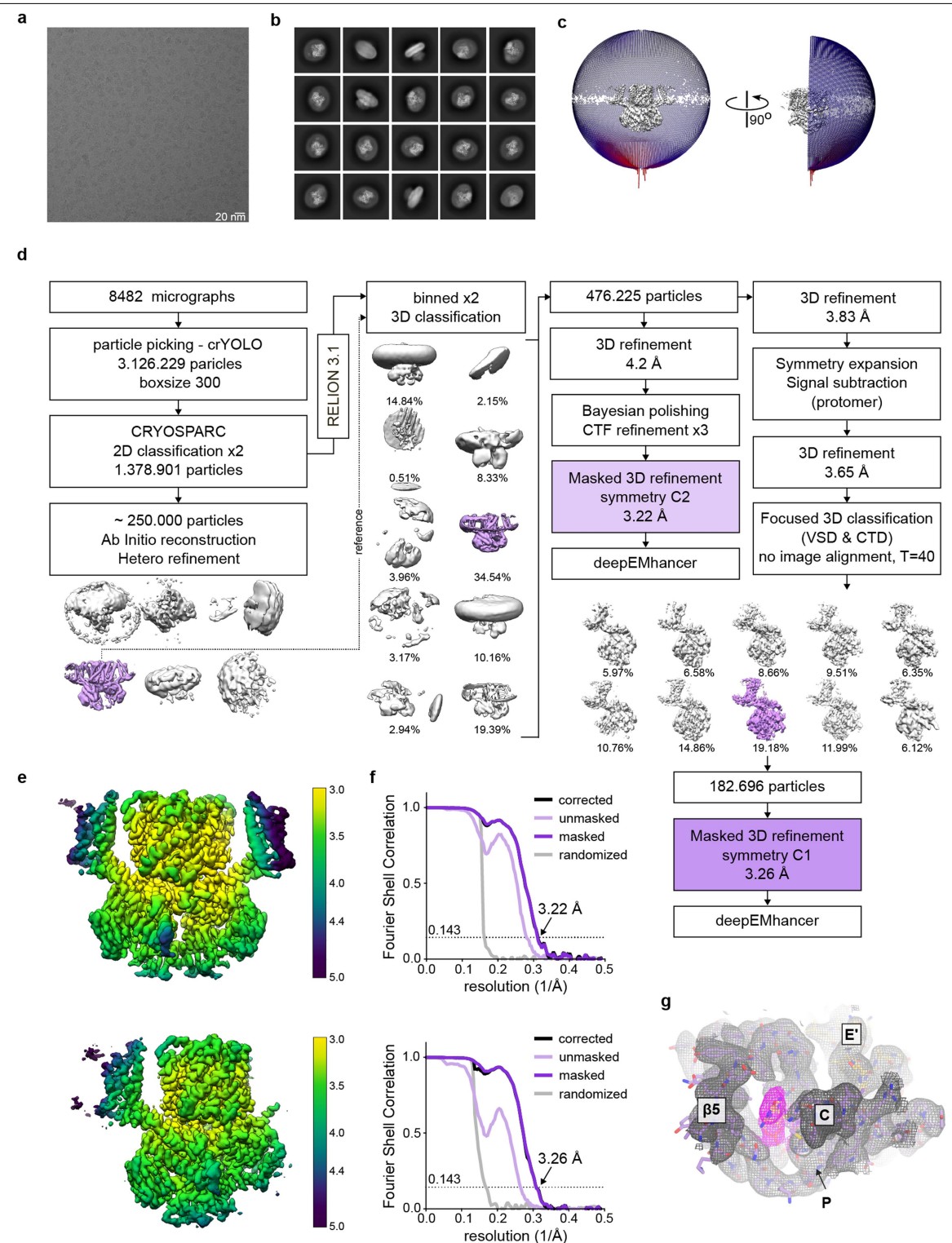

**Extended Data Fig. 7 | Cryo-EM processing workflow of SpSLC9C1 in lipid nanodiscs supplemented with cGMP.** Representative cryo-EM images out of 8,482 comparable images (a) and 2D classes (b) of vitrified SpSLC9C1 in presence of cGMP. c) Angular distribution of the obtained dimeric map. d) Image processing workflow. During 3D classification of protomers, the mask encompassing only the VSD and CTD was applied to focus on less-resolved parts of the protein. e) Final dimer and protomer deepEMhancer sharpened maps colored according to the local resolution, dimer map is displayed at 0.85 σ, protomer at 1.2 σ. f) Fourier Shell correlation plots of the obtained maps (0.143 criteria). g) cNMP-binding site of the CNBD from the final protomer map displayed as grey mesh contoured at 4.7 σ, protein model in purple, cGMP molecule in yellow, density corresponding to bound cGMP is shown in magenta.

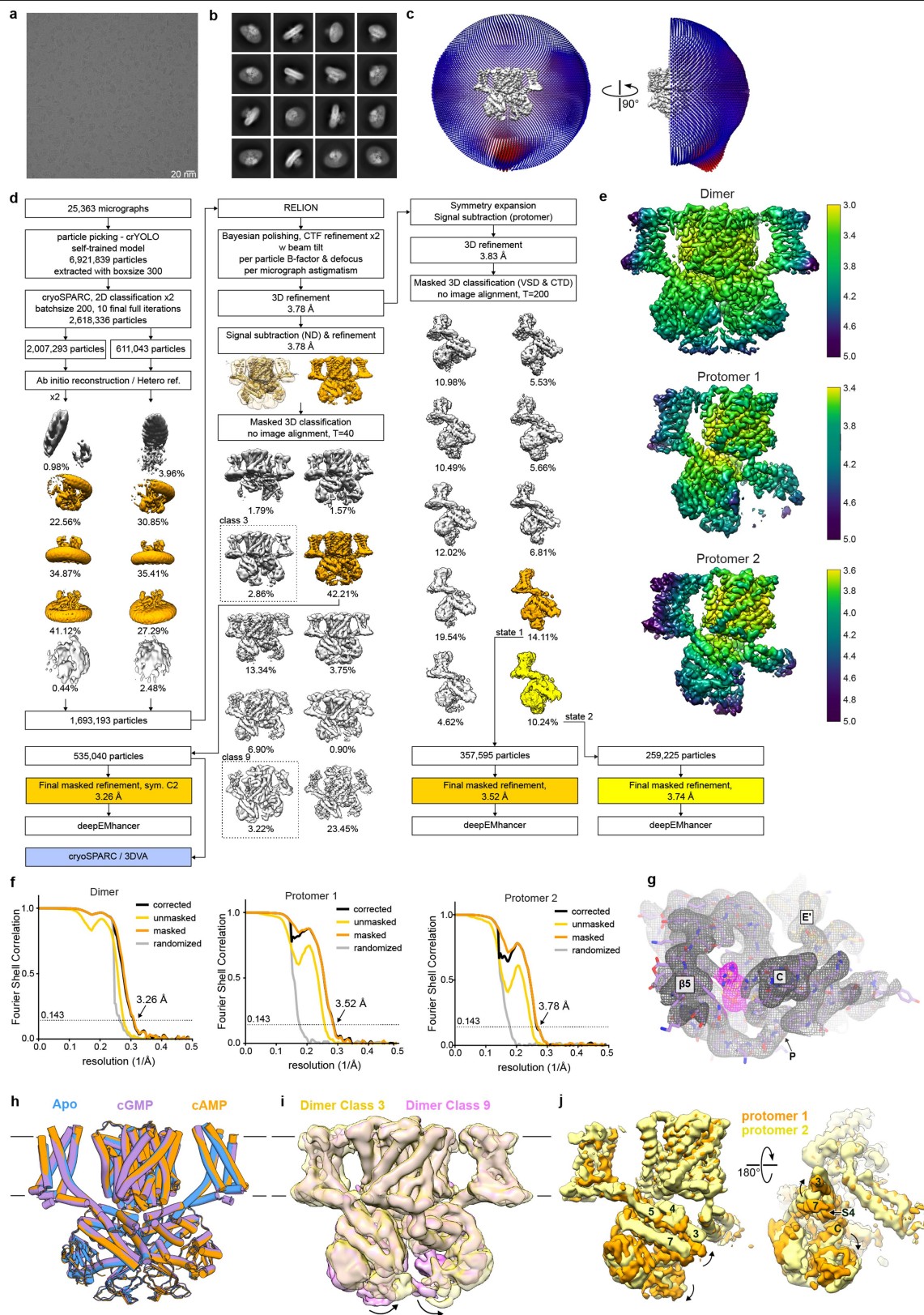

**Extended Data Fig. 8** | See next page for caption.

**Extended Data Fig. 8 | Cryo-EM processing workflow of SpSLC9C1 in lipid nanodiscs supplemented with cAMP.** Representative cryo-EM images out of 25,363 comparable images (a) and 2D classes (b) of vitrified SpSLC9C1 in presence of cAMP. c) Angular distribution of the best resolved dimeric class. d) Image processing workflow. Notably, the cytoplasmic domain displays a high degree of conformational heterogeneity throughout data processing, indicative of higher mobility. This is best exemplified by the intermediate processing step map at 3.78 Å resolution, by the various 3D classes obtained in the final 3D classification on a dimer and protomer level, as well as from the 3D variability analysis in CryoSparc (3DVA, see Supplementary Video 1). During 3D classification of protomers, the mask encompassing only the VSD and CTD was applied to focus on less-resolved parts of the protein. e) Final maps colored according to the local resolution, dimeric map is displayed at 0.5 σ, protomer 1 at 1 σ and protomer 2 at 0.8 σ, respectively. f) Fourier Shell Correlation plots (0.143 criteria) for the best dimer class (left), protomer 1 (middle) and protomer 2 (right) states. g) cNMP-binding site in the CNBD in protomer 1 map is displayed as grey mesh contoured at 4.7 σ, protein model is shown in purple, cAMP molecule in yellow, density corresponding to bound cAMP is displayed in magenta. h) Overlay of the best dimeric SpSLC9C1 structures in apo, cGMP-bound and cAMP-bound conformations. i) Overlay of the selected 3D classes (boxed in (d)), displaying the asymmetric movements of CTD between protomers. Class 3 is contoured at 1.3 σ and class 9 at 1.7 σ, respectively. j) Overlay of the two distinct protomer states displaying the largest conformational differences (see also Supplementary Video 2). Shown are refined masked maps contoured at 8.3 σ (protomer 1) and 8.4 σ (protomer 2).

**Extended Data Table 1 | Cryo-EM data collection and model validation statistics of apo SpSLC9C1**

| | Apo dimer (EMDB-17596) (PDB 8PCZ) | Apo protomer 1 (EMDB-17598) (PDB 8PD2) | Apo protomer 2 (EMDB-17599) (PDB 8PD3) | Apo protomer 3 (EMDB-17601) (PDB 8PD5) | Apo protomer 4 (EMDB-17602) (PDB 8PD7) | Apo GDN symmetric (EMDB-17604) | Apo GDN asymmetric (EMDB-17603) |
|---|---|---|---|---|---|---|---|
| | | | EMPIAR 11628 | | | EMPIAR 11629 | |
| **Data collection and processing** | | | | | | | |
| Magnification | 59,809 | 59,809 | 59,809 | 59,809 | 59,809 | 59,809 | 59,809 |
| Voltage (kV) | 300 | 300 | 300 | 300 | 300 | 300 | 300 |
| Electron exposure (e–/Å$^2$) | 60 | 60 | 60 | 60 | 60 | 60 | 60 |
| Defocus range (μm) | -0.3 to -3.0 | -0.3 to -3.0 | -0.3 to -3.0 | -0.3 to -3.0 | -0.3 to -3.0 | -0.3 to -3.0 | -0.3 to -3.0 |
| Pixel size (Å) | 0.836 | 0.836 | 0.836 | 0.836 | 0.836 | 0.836 | 0.836 |
| Symmetry imposed | C2 | C1 | C1 | C1 | C1 | C2 | C1 |
| Initial particle images (no.) | 5,022,107 | 5,022,107 | 5,022,107 | 5,022,107 | 5,022,107 | 3.982.593 | 3.982.593 |
| Final particle images (no.) | 1,458,809 | 509,751 | 379,058 | 202,420 | 241,549 | 683.441 | 598.677 |
| Map resolution (Å) | | | | | | | |
| FSC threshold | 3.21 | 3.25 | 3.3 | 3.34 | 3.34 | 3.05 | 3.3 |
| Map resolution range (Å) | 3.1 – 4.0 | 3.1 – 4.0 | 3.2 – 4.0 | 3.2 – 4.0 | 3.2 – 4.0 | 2.9 – 4.2 | 3.0 – 4.8 |
| | | | | | | | |
| **Refinement** | | | | | | | |
| Initial model used (PDB code) | 6Z3Y, 3J4Q | 8PCZ | 8PD2 | 8PD2 | 8PD2 | - | - |
| Model resolution (Å) | | | | | | | |
| FSC threshold | 3.0 | 3.1 | 3.1 | 3.1 | 3.1 | - | - |
| Model resolution range (Å) | 3.0 – 4.0 | 3.1 – 4.0 | 3.1 – 4.0 | 3.1 – 4.0 | 3.1 – 4.0 | - | - |
| Q-score | 0.43 | 0.47 | 0.46 | 0.44 | 0.44 | - | - |
| Model composition | | | | | | | |
| Non-hydrogen atoms | 16176 | 8270 | 8121 | 8093 | 8121 | - | - |
| Protein residues | 2046 | 1047 | 1028 | 1024 | 1028 | - | - |
| Ligands | - | | | - | - | - | - |
| *B* factors (Å$^2$) | | | | | | | |
| Protein | 137.83 | 113.16 | 114.99 | 117.79 | 111.12 | - | - |
| Ligand | - | - | - | - | - | - | - |
| R.m.s. deviations | | | | | | | |
| Bond lengths (Å) | 0.003 | 0.004 | 0.003 | 0.003 | 0.003 | - | - |
| Bond angles (°) | 0.559 | 0.520 | 0.560 | 0.534 | 0.592 | - | - |
| Validation | | | | | | | |
| MolProbity score | 1.38 | 1.09 | 1.17 | 1.03 | 1.14 | - | - |
| Clashscore | 5.81 | 3.02 | 3.87 | 2.18 | 3.44 | - | - |
| Poor rotamers (%) | 0.23 | 0 | 0 | 0.2 | 0.11 | - | - |
| Rama Z | | | | | | | |
| Whole | 1.86 | 0.72 | 0.70 | 1.83 | 1.39 | - | - |
| Helix | 1.85 | 0.83 | 0.81 | 1.80 | 1.29 | - | - |
| Sheet | 0.17 | -0.58 | 0.93 | 0.65 | 0.19 | - | - |
| Loop | -0.34 | -0.04 | -0.35 | -0.2 | 0.21 | - | - |
| Ramachandran plot | | | | | | | |
| Favored (%) | 97.73 | 98.55 | 98.72 | 97.83 | 98.52 | - | - |
| Allowed (%) | 2.27 | 1.45 | 1.28 | 2.17 | 1.48 | - | - |
| Disallowed (%) | 0 | 0 | 0 | 0 | 0 | - | - |

**Extended Data Table 2 | Cryo-EM data collection and model validation statistics of ligand-bound SpSLC9C1**

| | cGMP dimer (EMDB-17621) (PDB 8PDU) | cGMP protomer (EMDB-17622) (PDB 8PDV) | cAMP dimer (EMDB-17605) (PDB 8PD8) | cAMP protomer 1 (EMDB-17607) (PDB 8PD9) | cAMP protomer 2 (EMDB-17625) |
|---|---|---|---|---|---|
| | | EMPIAR 11630 | | EMPIAR 11635 | |
| **Data collection and processing** | | | | | |
| Magnification | 49,407 | 49,407 | 49,407 | 49,407 | 49,407 |
| Voltage (kV) | 200 | 200 | 200 | 200 | 200 |
| Electron exposure (e–/Å²) | ~51 | ~51 | ~51 | ~51 | ~51 |
| Defocus range (μm) | -0.3 to -2.0 | -0.3 to -2.0 | -0.3 to -2.0 | -0.3 to -2.0 | -0.3 to -2.0 |
| Pixel size (Å) | 1.022 | 1.022 | 1.022 | 1.022 | 1.022 |
| Symmetry imposed | C2 | C1 | C2 | C1 | C1 |
| Initial particle images (no.) | 3.126.229 | 3.126.229 | 6.921.839 | 6.921.839 | 6.921.839 |
| Final particle images (no.) | 476.225 | 182.696 | 535.040 | 357.595 | 259.225 |
| Map resolution (Å) | | | | | |
| FSC threshold | 3.22 | 3.26 | 3.26 | 3.52 | 3.78 |
| Map resolution range (Å) | 3.0 – 5.0 | 3.0 – 5.0 | 3.1 – 5.0 | 3.4 – 5.0 | 3.6 – 5.0 |
| | | | | | |
| **Refinement** | | | | | |
| Initial model used (PDB code) | 8PDV | 8PCZ | 8PD9 | 8PCZ | - |
| Model resolution (Å) | | | | | |
| FSC threshold | 2.7 | 2.9 | 2.9 | 3.3 | - |
| Model resolution range (Å) | 2.7 – 5.0 | 2.9 – 5.0 | 2.9 – 5.0 | 3.3 – 5.0 | - |
| Q-score | | | | | - |
| Model composition | | | | | |
| Non-hydrogen atoms | 16210 | 8140 | 16122 | 8107 | - |
| Protein residues | 2044 | 1027 | 2036 | 1024 | - |
| Ligands | 2 | 1 | 2 | 1 | - |
| | | | | | |
| *B* factors (Å²) | | | | | |
| Protein | 142.09 | 113.95 | 190.53 | 162.70 | - |
| Ligand | 196.96 | 131.84 | 282.53 | 183.05 | - |
| R.m.s. deviations | | | | | |
| Bond lengths (Å) | 0.002 | 0.003 | 0.005 | 0.005 | - |
| Bond angles (°) | 0.518 | 0.530 | 0.845 | 0.735 | - |
| Validation | | | | | |
| MolProbity score | 1.30 | 1.19 | 1.43 | 1.32 | - |
| Clashscore | 5.47 | 4.03 | 6.21 | 5.51 | - |
| Poor rotamers (%) | 0.2 | 0.2 | 0 | 0.5 | - |
| Rama Z | | | | | |
| Whole | 0.82 | 1.63 | 0.77 | 2.01 | - |
| Helix | 0.96 | 1.49 | 0.82 | 1.91 | - |
| Sheet | -0.31 | 0.62 | -0.02 | -0.90 | - |
| Loop | -0.36 | 0.12 | 0.04 | 0.37 | - |
| Ramachandran plot | | | | | |
| Favored (%) | 98.66 | 98.62 | 97.56 | 97.92 | - |
| Allowed (%) | 1.34 | 1.38 | 2.44 | 2.08 | - |
| Disallowed (%) | 0 | 0 | 0 | 0 | - |

# Reporting Summary

## Statistics

For all statistical analyses, confirm that the following items are present in the figure legend, table legend, main text, or Methods section.

| n/a | Confirmed | |
|---|---|---|
| ☐ | ☒ | The exact sample size (*n*) for each experimental group/condition, given as a discrete number and unit of measurement |
| ☐ | ☒ | A statement on whether measurements were taken from distinct samples or whether the same sample was measured repeatedly |
| ☒ | ☐ | The statistical test(s) used AND whether they are one- or two-sided *Only common tests should be described solely by name; describe more complex techniques in the Methods section.* |
| ☒ | ☐ | A description of all covariates tested |
| ☒ | ☐ | A description of any assumptions or corrections, such as tests of normality and adjustment for multiple comparisons |
| ☐ | ☒ | A full description of the statistical parameters including central tendency (e.g. means) or other basic estimates (e.g. regression coefficient) AND variation (e.g. standard deviation) or associated estimates of uncertainty (e.g. confidence intervals) |
| ☒ | ☐ | For null hypothesis testing, the test statistic (e.g. *F*, *t*, *r*) with confidence intervals, effect sizes, degrees of freedom and *P* value noted *Give P values as exact values whenever suitable.* |
| ☒ | ☐ | For Bayesian analysis, information on the choice of priors and Markov chain Monte Carlo settings |
| ☒ | ☐ | For hierarchical and complex designs, identification of the appropriate level for tests and full reporting of outcomes |
| ☒ | ☐ | Estimates of effect sizes (e.g. Cohen's *d*, Pearson's *r*), indicating how they were calculated |

*Our web collection on statistics for biologists contains articles on many of the points above.*

## Software and code

Policy information about availability of computer code

| Data collection | Cryo-EM: SerialEM 3.8.0 beta or 3.9.0 beta, EPU 2.7.0 or 2.8.0 (Thermo Fisher Scientific). ITC: MicroCal PEAQ-ITC Control and Analysis v1.41. NanoDSF: Prometheus Panta.Control 1.4.3 and Panta.Analysis 1.4.3 |
|---|---|
| Data analysis | Cryo-EM (managed through SBGrid version 2.5.6): cryoSPARC v3, Relion 3.1.0, Focus 1.1.0, MotionCor2 version 1.4.0, CTFfind4.1.14, crYOLO 1.7.5, 1.7.6 or 1.8.2, coot version 0.9.8.1, Phenix 1.20.1-4487, Isolde 1.6.0, deepEMhancer 20220530_cu10, Alphafold2 v2.3.2, Pymol 2.5.5, ChimeraX 1.6.1, Chimera 1.17.3 |

For manuscripts utilizing custom algorithms or software that are central to the research but not yet described in published literature, software must be made available to editors and reviewers. We strongly encourage code deposition in a community repository (e.g. GitHub). See the Nature Portfolio guidelines for submitting code & software for further information.

## Data

Policy information about availability of data

All manuscripts must include a data availability statement. This statement should provide the following information, where applicable:
- Accession codes, unique identifiers, or web links for publicly available datasets
- A description of any restrictions on data availability
- For clinical datasets or third party data, please ensure that the statement adheres to our policy

Data supporting the findings of this manuscript are available from the corresponding authors upon request. Cryo-EM density maps, half maps, and masks have been

# Research involving human participants, their data, or biological material

Policy information about studies with human participants or human data. See also policy information about sex, gender (identity/presentation), and sexual orientation and race, ethnicity and racism.

| | |
|---|---|
| Reporting on sex and gender | N/A |
| Reporting on race, ethnicity, or other socially relevant groupings | N/A |
| Population characteristics | N/A |
| Recruitment | N/A |
| Ethics oversight | N/A |

Note that full information on the approval of the study protocol must also be provided in the manuscript.

# Field-specific reporting

Please select the one below that is the best fit for your research. If you are not sure, read the appropriate sections before making your selection.

☒ Life sciences ☐ Behavioural & social sciences ☐ Ecological, evolutionary & environmental sciences

For a reference copy of the document with all sections, see nature.com/documents/nr-reporting-summary-flat.pdf

# Life sciences study design

All studies must disclose on these points even when the disclosure is negative.

| | |
|---|---|
| Sample size | No statistical methods were used to predetermine the sample size. Cryo-EM data collection was deemed sufficient if it was possible to classify the structural heterogeneity and to reach the resolution better than 4 Å in all of the subsequent 3D reconstructions originating from the same dataset. For ITC and nanoDSF experiments, the sample size was chosen according to the standard in the field. |
| Data exclusions | According to the standard in the cryo-EM field, micrographs with the resolution worse than 6 Å, displaying ice contamination/cracks, unusually high drift values, defocus values > 2 um for data collected at 200 kV, or > 3 um for the data collected at 300 kV, were excluded from the subsequent data analysis. Similarly, particles were excluded during 2D classification if they did not produce well-resolved classes with identifiable protein density, and during 3D classification, if they did not contribute to the 3D reconstructions with all of the protein domains resolved, and did not yield high-resolution 3D classes. For ITC and nanoDSF measurements, no data exclusion criteria were established prior to the data collection. |
| Replication | For ITC experiments, three biological replicates were measured. For nanoDSF experiments, three technical replicates were measured. All attempts at replication were successful. Protein was purified and reconstituted into nanodiscs >3 times with similar results. Structure determination was performed once per given dataset, for ligand-bound datasets several grids were imaged, producing micrographs of similar quality and appearance. |
| Randomization | Particles were randomized between even/odd groups during refinement and resolution estimation (gold-standard FSC). |
| Blinding | Blinding criteria are not applicable to cryo-EM data processing, as the data is handled in an automated fashion. |

# Reporting for specific materials, systems and methods

We require information from authors about some types of materials, experimental systems and methods used in many studies. Here, indicate whether each material, system or method listed is relevant to your study. If you are not sure if a list item applies to your research, read the appropriate section before selecting a response.

## Materials & experimental systems

| n/a | Involved in the study |
|-----|----------------------|
| ☒ ☐ | Antibodies |
| ☐ ☒ | Eukaryotic cell lines |
| ☒ ☐ | Palaeontology and archaeology |
| ☒ ☐ | Animals and other organisms |
| ☒ ☐ | Clinical data |
| ☒ ☐ | Dual use research of concern |
| ☒ ☐ | Plants |

## Methods

| n/a | Involved in the study |
|-----|----------------------|
| ☒ ☐ | ChIP-seq |
| ☒ ☐ | Flow cytometry |
| ☒ ☐ | MRI-based neuroimaging |

## Eukaryotic cell lines

Policy information about cell lines and Sex and Gender in Research

| | |
|---|---|
| Cell line source(s) | All of the cell lines used are commercially available: HEK293S GnTI- (CRL-3022, ATCC), Sf9 (12659017,ThermoFisher Scientific) |
| Authentication | No further authentication was performed after purchasing the cell lines |
| Mycoplasma contamination | All cell lines were tested for Mycoplasma every 3-4 months and were found negative |
| Commonly misidentified lines (See ICLAC register) | N/A |

