## [Peer Review File · Nature]

Manuscript Title: Structures of a sperm-specific solute carrier gated by voltage and cAMP

Reviewer Comments & Author Rebuttals

Reviewer Reports on the Initial Version:

Referee expertise:

Referee #1: VGICs, cryo-EM

Referee #2: NHE structure-function

Referees' comments:

Referee #1 (Remarks to the Author):

This is an interesting manuscript describing the first structures of the voltage-activated SLC9C1 Na/H antiporter. This unique antiporter is expressed in sperm flagella where it plays a key role in alkalinizing the cytoplasm and activating CatSper and driving chemotaxis. Previous sequence analysis and functional studies had shown that this transporter contains a functional S1-S4 voltage-sensing domain and intracellular cyclic nucleotide binding domain (CNBD) that allosterically control Na/H exchange, but the structural basis of these mechanisms remained unknown. Here, Kalienkova and colleagues present the first cryo-EM structures of this intriguing SLC9C1 transporter. A series of structures are presented for the *S. purpuratus* SLC9C1, for the apo protein, either in detergent and lipid nanodiscs at 0 mV, and then in detergent in the presence of cAMP or cGMP at lower resolution to provide a framework for how changes in voltage might control the transport cycle. The protein forms dimers, with two transport domains positioned at the dimer interface with the S1-S4-like domains in the periphery. The apo structures are similar and show how the S1-S4 domain forms a voltage-sensing domain with all the essential features known from extensive studies on related voltage-activated cation channels, but instead of coupling with a pore forming domain the S4 helix projects outside the membrane into the intracellular regions to insert into a tunnel formed between a series of intracellular helices and the CNBD. In these apo structures, the transport domains are in an inward facing conformation similar to previously seen in many other transporters adopting the NhaA fold with conserved Na binding sites deep within the TM regions and open to the intracellular side of the membrane. The dimer interface contains a hydrophobic cavity seen in related transporters and in the nanodisc structure an incompletely resolved lipid are bound near the interface. In previous functional studies cAMP diminished the extent of hyperpolarization needed to activate the transporter, so the authors reasoned that structures with cAMP bound at 0 mV might provide information on the nature of the conformational change involved in activation of the transporter. In the presence of cAMP and cGMP, density for the nucleotides are seen, but the accompanying conformational changes are small and of relatively unclear significance. Nevertheless, the authors propose that membrane hyperpolarization induces an inward movement of S4 to

diminish interactions between the intracellular domain and freeing the transport module to exchange Na and protons. In effect, the proposal is that inward movements of S4 diminish an inhibitory interaction between the cytoplasmic domain and the transport domain, thus enable the transport domain to exchange Na and protons. This mechanism is presented as an initial framework for thinking about the overall mechanism, but it is the apo structures that provide the most valuable foundation for understanding this hybrid family of cyclic nucleotide and voltage regulated transporters. Overall, the study appears to be of very high quality, the exposition clear and the conclusions nuanced and for the most part are appropriately cautious. The following are suggestions for improving the manuscript in revision.

1) It would be help for the authors to add figures highlighting the interactions between S4 and the CHs and CNBDs as this region is likely how movements of S4 unlock the transport domains. What kinds of residue interactions are present in this region and what might be said about how they would need to rearrange if we assume S4 moves perpendicular to the membrane by about 15-20 Å?

2) I had a hard time appreciating what was learned from the structures solved in the presence of cAMP or cGMP, and I think the figures could be improved to illustrate these better. Perhaps presenting local maps along with models in Fig. 4 would help? Also, what were the concentrations of cNMPs used in the structures? I could not find this information or how it relates to the concentrations used in ref 22.

3) The structure of the voltage-sensing domain in SLC9C1 is compared with cation channels containing CNBDs, which is reasonable, but most of the foundational work on voltage sensing comes from work on Shaker Kv channels and in places it might be worth making that connection. Ref 67 is important in identifying the charge transfer center in Shaker, but its not a substitute for many other important papers on that channel. A structure of Shaker is now available and the best estimates of charge per channel and which S4 residues contribute the most were done on that protein (PMIDs 8663993 and 8663992), for example. It would be nice to add structural comparisons with Shaker so that more of the foundational literature could be touched upon.

4) In maps in supplemental figures showing density corresponding to cAMP or cGMP it would help the reader if the additional density was highlighted with a different color.

5) The first sentence in the introduction is quite clunky and it would be good to revise.

Referee #2 (Remarks to the Author):

The study by Kalienkova et al. describes the cryo-EM-derived structure for the highly conserved and tissue-specialised sperm Na⁺/H⁺ exchanger from sea urchin *Strongylocentrotus purpuratus* (SpSLC9C1). This is a significant advancement in the ion transport field. Unlike the known structures

of homologs of the SLC9A and SLC9B subfamilies of the SLC9 superfamily of monovalent cation/protein exchangers as well as those of other secondary-active transporters, SLC9C1 and its close paralog SLC9C2 are predicted to contain a unique tripartite domain organization which includes not only a characteristic ion-translocating transmembrane domain (TMD), but also a membrane voltage-sensing domain (VSD) coupled to a cytoplasmic cyclic-nucleotide binding domain (CNBD) which preferentially favours cAMP.

In this report, the authors provide the first detailed glimpse of the 3D dimeric organization of SLC9C1 in an inward-facing conformation at 3.2 Å obtained using nanodisc-reconstituted protein. Unlike voltage-gated cyclic-nucleotide-regulated ion channels, the VSD is disconnected from the catalytic transport TMD and instead is positioned laterally and peripherally to the core of the protein where it is connected by a series of coupling helices (CH1-9) linked to the CNBD. The authors provide structural evidence supporting a novel gating-mechanism whereby membrane hyperpolarization (known to be initiated by egg-released speract peptide activation of a downstream sperm-resident K⁺ channel, SLO3) would likely cause a downward movement of the VSD positively-charged S4 helix which, in turn, would displace adjacent coupling helices and disrupt the cytoplasmic dimeric interfaces of the C-terminal domain containing the CNBD. It is postulated that these transpositions release the exchanger from a locked, inactive state. Previous studies have shown that activation of SLC9C1 causes sperm alkalization which stimulates soluble adenylylase (sAC) activity and cAMP production which, upon binding to the CNBD, lowers the barrier for voltage activation of SLC9C1. The authors provide some biophysical measurements (using differential scanning fluorimetry and isothermal titration calorimetry) indicating that in vitro binding of cAMP (and to a lesser extent cGMP) to purified CNBD stabilizes the domain structure, although how this stabilization affects voltage activation remains obscure. Concurrently, SLC9C1-induced alkalization also facilitates Ca²⁺ entry through the pH-sensitive Ca²⁺ channel CatSper and stimulation of sperm flagellar beat. This is critical as loss-of-function of SLC9C1 is known to be essential for sperm motility and fertilization and thus SLC9C1 is an attractive target for the development of male contraceptives.

Overall, the proposed molecular architecture for SLC9C1 is unequalled amongst ion transporters as it marries structural elements of a classic ion transporter with those found in voltage-gated cyclic-nucleotide-regulated ion channels. A shortcoming of this study is that it does not directly test some of the structural predictions of their model with mutagenesis and functional measurements. While this is not a critical deficiency as it would require considerable additional experimentation beyond the scope of the present study, the veracity of the derived structure remains tentative.

Notwithstanding, this is a commendable study that significantly advances our knowledge of this structurally and functionally diverse family of cation/proton exchangers, opens the door to further structural studies of SLC9C1, and thus should be of broad interest to the scientific community with potential pharmaceutical applications for managing human reproduction.

Other Comments:

1. Manuscript is original and well written. The structural data and interpretations appear sound and convincing.

2. Lines 161-163: The authors state that the TMD of SpSLC9C1, which was initially predicted to consist of 14 helices, instead comprises only 13 helices per protomer based on their structure.

However, the authors also indicate that the extreme amino terminus (residues 1-70)- illustrated as a dashed line in Fig. 1 - could not be resolved or modelled. This segment appears to contain a relatively hydrophobic sequence between amino acids 11-30 which in principle could be a signal sequence that is cleaved off or perhaps another transmembrane helix. Despite the reported structures of other Nha/NHE homologs that seemingly indicate only 13 TM helices (their N-termini were also difficult to model), how certain are the authors that there are only 13 helices? Previous biochemical studies (cysteine mutagenesis and biotin labelling) of human NHE1 indicated that the amino terminus was located intracellularly (rather than extracellularly) and that the transmembrane domain was comprised of 12 transmembrane helices plus an additional 2 intramembrane helices that formed a re-entrant loop (14 helices total) (see Wakabayashi, S., Pang, T., Su, X. & Shigekawa, M. (2000) A novel topology model of the human Na⁺/H⁺ exchanger isoform 1. *J. Biol. Chem* 275, 7942-7949). Please comment.

3. Lines 185-186: While the SpSLC9C1 possesses the conserved and catalytically critical 'ND' motif present in electroneutral Na⁺/H⁺ exchangers, this motif appears to be absent from mouse and human SLC9C1 and instead is replaced by a 'TS' motif. It would be informative to perform some structural modelling of the human SLC9C1 in this region and speculation on how this 'TS' motif might impact ion-binding and transport, if at all.

Author Rebuttals to Initial Comments:

We thank both reviewers for their very positive, considerate and constructive feedback! This makes science feel exciting and fun. We note that to accommodate to editorial policies we have already shortened the manuscript, yet making sure the main message is not lost. Below you find a detailed response to single comments.

Referee #1 (Remarks to the Author):

This is an interesting manuscript describing the first structures of the voltage-activated SLC9C1 Na/H antiporter. This unique antiporter is expressed in sperm flagella where it plays a key role in alkalinizing the cytoplasm and activating CatSper and driving chemotaxis. Previous sequence analysis and functional studies had shown that this transporter contains a functional S1-S4 voltage-sensing domain and intracellular cyclic nucleotide binding domain (CNBD) that allosterically control Na/H exchange, but the structural basis of these mechanisms remained unknown. Here, Kalienkova and colleagues present the first cryo-EM structures of this intriguing SLC9C1 transporter. A series of structures are presented for the *S. purpuratus* SLC9C1, for the apo protein, either in detergent and lipid nanodiscs at 0 mV, and then in detergent in the presence of cAMP or cGMP at lower resolution to provide a framework for how changes in voltage might control the transport cycle. The protein forms dimers, with two transport domains positioned at the dimer interface with the S1-S4-like domains in the periphery. The apo structures are similar and show how the S1-S4 domain forms a voltage-sensing domain with all the essential features known from extensive studies on related voltage-activated cation channels, but instead of coupling with a pore forming domain the S4 helix projects outside the membrane into the intracellular regions to insert into a tunnel formed between a series of intracellular helices and the CNBD. In these apo structures, the transport domains are in an inward facing conformation similar to previously seen in many other transporters adopting the NhaA fold with conserved Na binding sites deep within the TM regions and open to the intracellular side of the membrane. The dimer interface contains a hydrophobic cavity seen in related transporters and in the nanodisc structure an incompletely resolved lipid are bound near the interface. In previous functional studies cAMP diminished the extent of hyperpolarization needed to activate the transporter, so the authors reasoned that structures with cAMP bound at 0 mV might provide information on the nature of the conformational change involved in activation of the transporter. In the presence of cAMP and cGMP, density for the nucleotides are seen, but the accompanying conformational changes are small and of relatively unclear significance. Nevertheless, the authors propose that membrane hyperpolarization induces and inward movement of S4 to diminish interactions between the intracellular domain and freeing the transport module to exchange Na and protons. In effect, the proposal is that inward movements of S4 diminish an inhibitory interaction between the cytoplasmic domain and the transport domain, thus enable the transport domain to exchange Na and protons. This mechanism is presented as an initial framework for thinking about the overall mechanism, but it is the apo structures that provide the most valuable foundation for understanding this hybrid family of cyclic nucleotide and voltage regulated transporters. Overall, the study appears to be of very high

quality, the exposition clear and the conclusions nuanced and for the most part are appropriately cautious. The following are suggestions for improving the manuscript in revision.

1) It would be help for the authors to add figures highlighting the interactions between S4 and the CHs and CNBDs as this region is likely how movements of S4 unlock the transport domains. What kinds of residue interactions are present in this region and what might be said about how they would need to rearrange if we assume S4 moves perpendicular to the membrane by about 15-20 Å?

We would like to thank the reviewer for this suggestion. We have generated additional figures displaying the interactions between S4 and the coupling helices, as well as between S4 and the CNBD (included in Extended Data Figure 6c-f). These are mostly composed of hydrophobic surface interactions, which make it hard to pinpoint a specific interaction that would have to be disrupted or formed due to movements of S4 upon hyperpolarisation. We have thus kept it more descriptive and added the following to the main text:

- *Lines 153-154: “The interactions between S4 and the coupling helices appear to be mediated by numerous hydrophobic residues (Extended Data Fig. 6d-f).”*
- *Lines 166-168: “As a consequence, the nucleotide binding site is facing the symmetry axis of the molecule (Fig. 3b), and helices C and D come in direct contact with the cytoplasmic tip of VSD’s S4 (Extended Data Fig. 6f).”*
- *Lines 262-267: “They are accompanied by subtle conformational transitions in the VSD and CNBD, further indicating that the CHs might play a central role in coupling voltage-sensing and cAMP modulation to exchange activity in the TD. In line with this assumption, the extended cytoplasmic region of S4 forms several interaction areas with CHs, which have to rearrange upon voltage-activation (Extended Data Fig. 6d,e).”*

2) I had a hard time appreciating what was learned from the structures solved in the presence of cAMP or cGMP, and I think the figures could be improved to illustrate these better. Perhaps presenting local maps along with models in Fig. 4 would help? Also, what were the concentrations of cNMPs used in the structures? I could not find this information or how it relates to the concentrations used in ref 22.

Thanks for raising this point. The most evident observation is the higher degree of flexibility in the cytoplasmic domains seen upon cAMP addition. Since this is mostly characterized by a less well resolved cryo-EM density in the CTD, it is somewhat more challenging to visualize (now highlighted by arrows in Figure 4a). We further performed a 3D variability analysis (3DVA) on the dimeric cAMP-bound SpSLC9C1 data with nanodisc density subtracted (described in Methods and in Extended Data Figure 9) and illustrate the observed variability in a new Supplementary Video 1. To highlight better the largest conformational changes observed upon cAMP addition, as seen from the two distinct

protomer states obtained, we added a morph between the two respective protomeric cryo-EM maps (Supplementary Video 2) and display these in a new panel in Extended Data Figure 9i.

We have revised the main text at several instances to describe the obtained conformational heterogeneity better, such as:

- Lines 180-207: “To investigate the impact of ligand binding on the conformation of SpSLC9C1, we solved the structures of the protein in presence of cGMP and cAMP (Extended Data Fig. 8 and 9 and Extended Data Table 2). To resolve the structural heterogeneity, we performed focused classifications and refinements on a dimer and protomer level for both ligand-bound datasets. In all cases, we could identify prominent densities within the CNBD corresponding to bound nucleotides, which were not present in the apo cryo-EM maps (Extended Data Fig. 1h, 8g and 9g). Overall, the best resolved cGMP- and cAMP-bound dimer structures, at 3.2 Å and 3.3 Å, respectively, share a similar conformation to the apo structure (Extended Data Fig. 9h). Yet, we could identify key differences. While image processing of the cGMP dataset revealed only one dominant conformation, the cAMP-bound dataset displayed a high degree of conformational heterogeneity throughout data analysis, where in particular the CTD was generally less resolved and mobile (Fig. 4a, Extended Data Fig. 8d,e and 9b,d,e,i). In most of the 3D classes, and as observed in the 3D variability analysis, the dimer interaction between the β -CTDs is disrupted in presence of cAMP, causing the CTD to swing away from the symmetry axis (Fig. 4a, Extended Data Fig. 9d,i and Supplementary Video 1). The observed flexibility in the CTD is also accompanied by a higher mobility of the VSDs, although at no point does the VSD approach the transport domain. Further, the extracellular tip of TM1 rotates around P75 towards the symmetry axis, pushing the TD protomers apart (Extended Data Fig. 7e). The observed movement is similar to the ‘breathing motions’ described for NHE9¹⁷, underlining the dynamic nature of the TD dimer interface as a common feature in the SLC9 family. On a protomer level, two distinct classes could be resolved that largely resemble conformational changes observed on a dimer level, and which are best appreciated in a morph between both cryo-EM maps (Extended Data Fig. 9d-f,j and Supplementary Video 2). Sufficiently resolved regions further allow to describe in more detail the conformational heterogeneity induced by cAMP-binding. Here, we observed pronounced movements of in particular CH3 and CH4, as well as CH7 and S4 upwards, away from the CNBD, and the downward movement of CNBD, best resolved for helix C (Fig. 4d and Supplementary Video 2). “
- Lines 217-226: “Secondly, we observe a high degree of conformational heterogeneity in the CTDs upon cAMP addition. Therefore, we hypothesize that a full conformational transition of the C-helix upon cAMP binding will require further rearrangements within the cytoplasmic domain. This is supported by the presumably ‘initial’ transitions observed in the cAMP-bound protomer state 2 (Fig. 4d, Extended Data Fig. 9j and Supplementary Video 2). While in the apo structures, the C- and D- helices are in direct contact with S4, and might preclude its movement, upon cAMP binding S4 and the surrounding CHs move

upwards, closer to the membrane, and the C-helix moves downwards, further away from S4, possibly releasing the inhibition. As a result, the β -CNBD might be approaching the C-helix reminiscent of other ligand-bound CNBD structures.”

- *Lines 291-298: “Secondly, the CHs act as an allosteric transducer, by mediating conformational changes and thereby coupling all three functional units. The cAMP-bound structures provide a glimpse into the mechanism of modulation of SpSLC9C1 by cNMPs and putative coupling between the functional domains via the CHs. In general, movements of the CHs are linked to subtle rearrangements in the VSD and CNBD, and binding of its strong agonist cAMP induces large conformational transitions within the CTD (Fig. 4a, Extended Data Fig. 2f, 9j and Supplementary Video 1,2). ”*

With respect to the cNMP concentration, we apologize for this omission. We have now added the following statement in the materials and methods, line 793: “Ligands (cAMP or cGMP) were added shortly before freezing to a final concentration of 2 mM.” The authors of the functional study (Windler et al., 2018, PMID: 30022052) included 1 mM cNMPs in the pipette solution for patch clamp experiments, which is in a similar range as the concentration we used for cryo-EM.

3) The structure of the voltage-sensing domain in SLC9C1 is compared with cation channels containing CNBDs, which is reasonable, but most of the foundational work on voltage sensing comes from work on Shaker Kv channels and in places it might be worth making that connection. Ref 67 is important in identifying the charge transfer center in Shaker, but its not a substitute for many other important papers on that channel. A structure of Shaker is now available and the best estimates of charge per channel and which S4 residues contribute the most were done on that protein (PMIDs 8663993 and 8663992), for example. It would be nice to add structural comparisons with Shaker so that more of the foundational literature could be touched upon.

Indeed, as the reviewer pointed out, we have mostly restricted our comparison to hyperpolarization-activated channels, and some of the other CNBD channels of known structure. Yet, we acknowledge the reviewer’s suggestion and have included for comparison the sequence and the structure of the Shaker channel’s VSD in Extended data figure 6a,b. Further, we now indicate that gating charges numbering is according to Shaker nomenclature in figure legends. Overall we have included the references the reviewer suggested and additionally PMID 8789953 and included the following statements in the main text:

- *Lines 134-147: “The VSD of SpSLC9C1 rather resembles that of the depolarization-activated Shaker Kv channel³⁵, including a short amphipathic helix S0 (Extended Data Fig. 6b). SpSLC9C1 has 7 positively charged residues in a canonical (R/K-XX)_n pattern on S4, which almost matches the number of those in HCN1³⁰ and HCN4³¹ (Fig. 3c and Extended Data Fig. 6a). The voltage-sensing helix S4 is in the inactive upward position, as no activating hyperpolarizing conditions can be established in detergent or nanodiscs. All*

positively charged residues reside within the membrane and R₁ (R803), which is homologous to R368 in Shaker Kv and is supposed to cross the entire membrane electric field upon activation^{1,36-38}, is still high above the gating charge transfer centre (GCTC) (Fig. 3c). Interestingly, the GCTC of SpSLC9C1 and that of other SLC9C1 homologs, has a Tyr rather than a Phe, as found in the majority of VSDs^{39,40} (Extended Data Fig. 4a and 6a). Yet, the VSD of SpSLC9C1 was shown to be a functional voltage sensor¹, in line with extensive mutagenesis studies on Shaker Kv channel, which concluded that only Tyr and Trp substitutions preserved the WT-like channel activation at more negative membrane potential⁴⁰.”

4) In maps in supplemental figures showing density corresponding to cAMP or cGMP it would help the reader if the additional density was highlighted with a different color.

We have colored the cNMP density differently.

5) The first sentence in the introduction is quite clunky and it would be good to revise.

We have revised the first sentence.

Referee #2 (Remarks to the Author):

The study by Kalienkova et al. describes the cryo-EM-derived structure for the highly conserved and tissue-specialised sperm Na⁺/H⁺ exchanger from sea urchin *Strongylocentrotus purpuratus* (SpSLC9C1). This is a significant advancement in the ion transport field. Unlike the known structures of homologs of the SLC9A and SLC9B subfamilies of the SLC9 superfamily of monovalent cation/protein exchangers as well as those of other secondary-active transporters, SLC9C1 and its close paralog SLC9C2 are predicted to contain a unique tripartite domain organization which includes not only a characteristic ion-translocating transmembrane domain (TMD), but also a membrane voltage-sensing domain (VSD) coupled to a cytoplasmic cyclic-nucleotide binding domain (CNBD) which preferentially favours cAMP.

In this report, the authors provide the first detailed glimpse of the 3D dimeric organization of SLC9C1 in an inward-facing conformation at 3.2 Å obtained using nanodisc-reconstituted protein. Unlike voltage-gated cyclic-nucleotide-regulated ion channels, the VSD is disconnected from the catalytic transport TMD and instead is positioned laterally and peripherally to the core of the protein where it is connected by a series of coupling helices (CH1-9) linked to the CNBD. The authors provide structural

evidence supporting a novel gating-mechanism whereby membrane hyperpolarization (known to be initiated by egg-released speract peptide activation of a downstream sperm-resident K⁺ channel, SLO3) would likely cause a downward movement of the VSD positively-charged S4 helix which, in turn, would displace adjacent coupling helices and disrupt the cytoplasmic dimeric interfaces of the C-terminal domain containing the CNBD. It is postulated that these transpositions release the exchanger from a locked, inactive state. Previous studies have shown that activation of SLC9C1 causes sperm alkalinization which stimulates soluble adenylylase (sAC) activity and cAMP production which, upon binding to the CNBD, lowers the barrier for voltage activation of SLC9C1. The authors provide some biophysical measurements (using differential scanning fluorimetry and isothermal titration calorimetry) indicating that in vitro binding of cAMP (and to a lesser extent cGMP) to purified CNBD stabilizes the domain structure, although how this stabilization affects voltage activation remains obscure. Concurrently, SLC9C1-induced alkalinization also facilitates Ca²⁺ entry through the pH-sensitive Ca²⁺ channel CatSper and stimulation of sperm flagellar beat. This is critical as loss-of-function of SLC9C1 is known to be essential for sperm motility and fertilization and thus SLC9C1 is an attractive target for the development of male contraceptives.

Overall, the proposed molecular architecture for SLC9C1 is unequalled amongst ion transporters as it marries structural elements of a classic ion transporter with those found in voltage-gated cyclic-nucleotide-regulated ion channels. A shortcoming of this study is that it does not directly test some of the structural predictions of their model with mutagenesis and functional measurements. While this is not a critical deficiency as it would require considerable additional experimentation beyond the scope of the present study, the veracity of the derived structure remains tentative. Notwithstanding, this is a commendable study that significantly advances our knowledge of this structurally and functionally diverse family of cation/proton exchangers, opens the door to further structural studies of SLC9C1, and thus should be of broad interest to the scientific community with potential pharmaceutical applications for managing human reproduction.

Other Comments:

1. Manuscript is original and well written. The structural data and interpretations appear sound and convincing.

Thanks!

2. Lines 161-163: The authors state that the TMD of SpSLC9C1, which was initially predicted to consist of 14 helices, instead comprises only 13 helices per protomer based on their structure. However, the authors also indicate that the extreme amino terminus (residues 1-70)- illustrated as a dashed line in Fig. 1 - could not be resolved or modelled. This segment appears to contain a relatively hydrophobic sequence between amino acids 11-30 which in principle could be a signal sequence that is cleaved off or perhaps another transmembrane helix. Despite the reported structures of other Nha/NHE

homologs that seemingly indicate only 13 TM helices (their N-termini were also difficult to model), how certain are the authors that there are only 13 helices? Previous biochemical studies (cysteine mutagenesis and biotin labelling) of human NHE1 indicated that the amino terminus was located intracellularly (rather than extracellularly) and that the transmembrane domain was comprised of 12 transmembrane helices plus an additional 2 intramembrane helices that formed a re-entrant loop (14 helices total) (see Wakabayashi, S., Pang, T., Su, X. & Shigekawa, M. (2000) A novel topology model of the human Na⁺/H⁺ exchanger isoform 1. *J. Biol. Chem* 275, 7942-7949). Please comment.

*The reviewer rightly points out that there is a hydrophobic sequence in SpSLC9C1 N-terminus, which was not resolved in our structures. It is likely that due to hydrophobicity of this region it was predicted to be membrane-embedded, also by AlphaFold (<https://alphafold.ebi.ac.uk/entry/A0A7M7T0D5>). However, if that was the case, we would expect to observe at least some low-resolution helical density in the vicinity of the transport domain, or even as part of the dimer interface, as seen in a recent structure of NHA2 (Matsuoka et al., *NSMB* 2022, PMID 35173351). This is not the case and we do not observe anything resembling an additional N-terminal transmembrane helix. With respect to the NHE1 biochemical study, if we compare the predicted topology with the recent NHE1 structure (Dong et al., *Nature Communications* 2021, PMID 34108458) – some of the topology was predicted correctly, while some of it wasn't. Undoubtedly valuable information can be obtained by biochemical accessibility experiments, however, the results are not always reliable and in particular negative results not entirely conclusive. We do observe a poorly resolved density forming 'a cap' above the dimer interface, which is seen best at lower contour (e.g., Extended data figure 5a), and which we indicate to potentially be attributed to the N-terminus (to tone this statement down it was now moved from the main text to the figure legend). To investigate if the N-terminus might have been cleaved off, we performed mass spectrometry analysis on the purified protein (band excised from the gel). The analysis confirms the presence of amino-acids 10–47 in our sample (please see screenshot below), indicating that this region is probably not cleaved off and is not a signal peptide, at least not in HEK cells which we used as expression system.*

However, since at present we cannot absolutely exclude the existence of an additional N-terminal transmembrane helix, we have rephrased the sentence to purely describe our observation:

- *Lines 101-104: "While predicted to consist of 14 TMs¹, we could only identify 13 resolved helices per protomer, with the first 70 N-terminal residues unresolved (Fig. 2a,b and Extended Data Fig. 4a)."*

Proteins									
1	spSLC9C1	497.95		72%	72%	336	336		14745
2	tr J3QS39 J3QS39_HU...	133.84		51%	51%	4	4		1046
3	P68363 TBA1B_HUMAN	233.67		47%	47%	14	14		5015
4	tr Q55T81 Q55T81_HU...	215.91		35%	35%	10	2		4174
5	O43707 ACTN4_HUMAN	264.92		31%	31%	21	13		10485
6	P68371 TBB4B_HUMAN	199.86		28%	28%	9	0		4983
7	tr A0A0G2JIW1 A0A0G...	192.46		26%	26%	12	9		7010
8	P11142 HSP7C_HUMAN	220.61		26%	26%	11	9		7089
9	P04264 K2C1_HUMAN	210.19		23%	23%	11	9		6603
10	P13645 K1C10_HUMAN	188.80		22%	22%	8	8		5882
ows spectrum only found by denovo N		178.83		21%	21%	7	1		5043

Coverage	Peptides	Denovo Only Tags
>spSLC9C1		
1	MSKRRVVKLR	ELVPAVAALA VAVLIQSATG SSSGSGHTPT TQATHAD

3. Lines 185-186: While the SpSLC9C1 possesses the conserved and catalytically critical 'ND' motif present in electroneutral Na⁺/H⁺ exchangers, this motif appears to be absent from mouse and human SLC9C1 and instead is replaced by a 'TS' motif. It would be informative to perform some structural modelling of the human SLC9C1 in this region and speculation on how this 'TS' motif might impact ion-binding and transport, if at all.

This is a very puzzling feature of mammalian SLC9C1 homologs, as most of them, if not all, lack this critical motif. At present it is controversial if mammalian SLC9C1s can even function as Na⁺/H⁺ exchangers. However, it is undisputed that these proteins are essential for sperm motility, as shown for mice and humans. We are currently investigating the structural differences between SpSLC9C1 and its mammalian counterparts, and the peculiarities in terms of function. However, at present we do not know enough to speculate what the biological role of these proteins in mammals might be. We refer to this in our last discussion paragraph.

- *Lines 318-326: "Hence, SLC9C1 is a potential target for the treatment of male infertility, as well as for non-hormonal on-demand male contraceptives, analogously to a recent strategy proposed for sAC⁵⁶. However, the exact function of mammalian SLC9C1s in sperm capacitation remains unclear, as they lack conserved residues at the cation-binding site and mice SLC9C1^{-/-} sperm display a defective cAMP signalling, rather than an impaired pH_i regulation^{2,11,57-59}. This questions and the linked pharmacological potential would benefit from further characterization of mammalian SLC9C1, including how mammalian SLC9C1s differ from SpSLC9C1 functionally and structurally."*

Reviewer Reports on the First Revision:

Referees' comments:

Referee #1:

The authors have done an excellent job of revising the manuscript to address the suggestions of the reviewers. I have no further comments and think that in its current form the manuscript is ready and appropriate for publication in Nature.

Referee #2:

The authors have satisfactorily addressed the comments from the initial review.